# A cross-border seroprevalence study on HBV, HCV, HDV and HIV in remote Amazonian communities on the border between French Guiana and Suriname

Roxane Schaub[1,2☯*], M. Sigrid MacDonald–Ottevanger[3,4☯], Stella Hoang[1,5], Amandine Pisoni[6], Karine Bolloré[6], Barbara Biche[1], Anfernee Neus[7], Rikesh Bisnajak[8], Janke Schinkel[4], Julie Blanc[1], Antoon Grunberg[3], Richard Naldjinan[9], Aude Lucarelli[10], Cyril Rousseau[11], Céline Michaud[11], Melanie Gaillet[11,12], Soeradj Harkisoen[8,13], Emmanuel Gordien[14], Maria Prins[15,16], Edouard Tuaillon[6], Mathieu Nacher[1,2☯], Stephen Vreden[3☯]

1 CIC Inserm 1424, Centre Hospitalier Universitaire de Guyane, Cayenne, French Guiana, 2 Inserm UA 17 Santé des Populations en Amazonie, Cayenne, French Guiana, 3 Foundation for Scientific Research Suriname (SWOS), Paramaribo, Suriname, 4 Department of Medical Microbiology, Amsterdam University Medical Center, Amsterdam, The Netherlands, 5 Service de Maladies Infectieuses et Tropicales, CHU de la Réunion, St Denis, La Réunion, 6 Pathogenesis and Control of Chronic and Emerging Infections, University of Montpellier, INSERM, EFS, Antilles University, Montpellier University Hospital, Montpellier, France, 7 Faculty of Medicine, Anton de Kom University, Paramaribo, Suriname, 8 Department of Internal Medicine, Academic Hospital, Paramaribo, Suriname, 9 Unité des Maladies Infectieuses et Tropicales, Centre Hospitalier Universitaire de Guyane, Cayenne, French Guiana, 10 COREVIH Guyane, Institut Santé des Populations en Amazonie, Centre Hospitalier Universitaire de Guyane, Cayenne, French Guiana, 11 Centres Délocalisés de Prévention et de Soins, Centre Hospitalier Universitaire de Guyane, Cayenne, French Guiana, 12 Laboratoire TIMC-IMAG, UMR 5525 CNRS, Université Grenoble Alpes, Grenoble, France, 13 Department of Medical Microbiology, University Medical Center Utrecht, Utrecht, The Netherlands, 14 Clinical Microbiology Laboratory, National Reference Centre for Hepatitis B, C and Delta Viruses, Hôpital Avicenne Assistance Publique-Hôpitaux de Paris, Bobigny, France, 15 Public Health Service of Amsterdam, Department of Infectious Diseases, Amsterdam, The Netherlands, 16 Amsterdam Infection and Immunity (AII), Department of Infectious Diseases, Amsterdam UMC, University of Amsterdam, Amsterdam, The Netherlands

☯ These authors contributed equally to this work.
* roxane.schaub@ch-cayenne.fr

## Abstract

### Background

National epidemiological data are essential to achieve viral hepatitis elimination goals. However, viral hepatitis and HIV epidemiology in French Guiana (FG) and Suriname is mainly limited to urban areas. We therefore assessed the prevalence, associated determinants, and knowledge, attitudes, practices, and beliefs (KAP-B) regarding hepatitis B (HBV), hepatitis C (HCV), hepatitis D (HDV), and HIV in the isolated area of the Maroni River, bordering FG and Suriname.

**Data availability statement:** Data cannot be shared publicly due to General Data Protection Regulation (GDPR) concerning so-called sensitive health data. Data is available from the promoter (Centre Hospitalier Universitaire de Guyane; contact via the CHU de Guyane Promotion Unit) for researchers who meet the criteria for access to sensitive data.

**Funding:** The MAHEVI project was funded by the European Regional Development Fund (Programme de Coopération Interreg Amazonie FEDER/2016/N°189 N° Synergie-CTE 3918) awarded to MN and by the Agence Nationale de Recherche sur le VIH/SIDA et les Hépatites virales (ANRS 95025) awarded to MN. The funder had no role in study design, data collection and analysis, and in the writing of this manuscript.

**Competing interests:** The authors have declared that no competing interests exist.

## Methods

Between January 2018 and February 2019, adults (≥18 years) living along the Maroni were enrolled in this cross-sectional study and completed a risk factor and KAP-B questionnaire. Participants were tested for viral hepatitis and HIV using serological and molecular analyses: HBV (HBsAg, anti-HBcore antibodies, anti-HBs, HBV DNA), HCV (HCV-antibodies, HCV RNA), HDV (HDV-antibodies), and HIV (HIV-antibodies/ antigen, HIV RNA). Age- and sex-adjusted HBV, HCV, and HIV prevalences were estimated using post-stratification weighting.

## Findings

Among 2286 participants, adjusted prevalences were respectively 2.08% (95%CI: 1.49–2.66; n = 46) for HBsAg, 25.6%, (95%CI: 20.89–29.24, n = 646) for resolved HBV infection (negative HBsAg and positive anti-HBcore antibodies), 0.13% (95%CI: 0.0–0.27; n = 5) for HCV-antibodies, and 0.65% (95%CI: 0.0–1.40; n = 12) for HIV-antibodies. There were no HDV infections. HBV exposure (acute or resolved) was independently associated with age (per 10-year increase aOR:1.27, 95%CI: 1.18–1.37, p < 0.0001), males (aOR:1.34, 95%CI: 1.10–1.65, p = 0.006), Maroon (aOR:2.22, 95%CI: 1.27–4.06) or other ethnic groups (aOR:2.94, 95%CI: 1.61–5.59) versus Amerindians, (p = 0.002), and higher education (aOR:0.71, 95%CI: 0.56–0.90, p = 0.005). Older age (per 10-years increase aOR:1.64, 95%CI: 1.11–2.44), p = 0.01) and non-autochthonous origin (aOR:3.81, 95%CI: 1.14–12.26, p = 0.02) were associated with HIV. Less than one in six participants correctly identified viral hepatitis transmission modes.

## Conclusion

HBV, HCV and HIV prevalences along the Maroni River were comparable to urban areas and remain concerning. Knowledge about viral hepatitis was low. Achieving viral hepatitis elimination goals requires awareness campaigns and test-and-treat strategies tailored to hard-to-serve populations.

## Introduction

The HIV epidemic has led to thousands of cumulative deaths on the Guiana Shield, where HIV was introduced in the 1970s and soared in the 1990s with multidirectional circulation [1]. French Guiana (FG) and Suriname, both situated on the Guiana Shield, have similar HIV prevalence estimates: 1.2%−1.4% of the total population in 2016 [2] and 1.6% (95% confidence interval (95%CI): 1.3–1.9), respectively, in adults aged 15–49 years in 2022 [3]. Transmission is predominantly heterosexual (87.7%), followed by unspecified (8.7%) and mother-to-child transmission (MTCT, 1.9%) [4], AIDS and HIV-related mortality have, however, substantially decreased, most likely as a result of increased availability of combination antiretroviral therapy (cART), early treatment initiation, and upscaling of preventive measures campaigns (HIV

screening, Treatment as Prevention (TasP), Pre-Exposure Prophylaxis (PrEP) awareness, condom-use). Recent efforts have broadened the focus of HIV programs towards sexual health and sexually transmitted infections [5], and prevention of mother-to- child transmission, notably including hepatitis B (HBV) [6].

Countries on the Guiana Shield share a setting wherein most of the population lives in urban areas along a narrow coastal strip while the rest lives in scattered small villages in the vast interior, most of which is covered by Amazonian forest. Both countries also share a richness in gold in the interior, which attracts tens of thousands of mostly undocumented, artisanal gold miners with a particular social organization around the mines, notably sex work [7]. Interestingly, on the rural border region between French Guiana and Suriname, the Maroni, a recent study showed a 0.52% HIV prevalence (95%CI: 0.45–0.59), far lower than the overall HIV prevalence in both countries. Given the paucity of information on the Maroni, it is unknown if this variation in prevalence is also seen in viral hepatitis. In the more urban areas of FG and Suriname, HBV prevalence oscillates around 3%, with varying prevalence across genders and ethnic groups [8–11] and HCV prevalence varies between 0.2% to 1.1% in FG [9] and is estimated at 1.0% in Suriname [12], with significant differences in gender and ethnic groups. However, a recent study in rural villages in the north east of Suriname, showed a much higher HCV prevalence of 5.9% (95%CI: 1.5–10.3) [10]. These variations emphasize the need to conduct epidemiological research nationwide in order to fully comprehend the dynamics of a disease, including differences in populations and geographical areas, and the influence of cultural customs. We therefore aimed to assess viral hepatitis in a large rural border region between Suriname and French Guiana.

Apart from the towns of Albina (5,247 inhabitants) and Saint-Laurent-du-Maroni (45,000 inhabitants), the 520 km long border between Suriname and FG is mostly unenforced (Fig 1), and consists of Amazonian forest where villages on either side of the Maroni River can only be reached by canoe or small aircraft. Most of the population consists of Maroon tribes -descendants of West-Africans, who escaped enslavement-, and indigenous tribes. The population of Eastern Suriname is around 31,000 persons and that of Western FG is around 90,000 persons and growing rapidly [4]. The border river is mostly symbolic, with relatives living on both shores, leading to frequent cross-border migration and travel to Saint-Laurent-du-Maroni, the main town in the Maroni basin [13]. In this urban hub of Western FG, the prevalence of HBsAg seropositivity between 2012 and 2016 was 2.1%; 1.1% in women and 3.7% in men [4].

In this remote area, health care is provided through health clinics in the main villages (Fig 1), some sites staffed by doctors whereas others are staffed by nurses [4]. Knowledge, attitudes, practices and beliefs (KAP-B) surveys have shown a high prevalence of high-risk sexual behavior, and infrequent sexual interactions between Amerindians and Maroon communities [15]. In the 1990s, the HIV epidemic surged from zero to over 1% prevalence among pregnant women on the Maroni [16]. MTCT is an estimated 0.7% between 2013–2022 in FG [16] and 1.8% in 2014 in Suriname [6]. This, coupled with various cultural practices, including subcutaneous penile implants (*bugrus*) [17], vaginal steam baths -which can lead to disruption of mucosal integrity [18] and a matriarchal society with migrant and sometimes polygamous men, led to the perception that prevalence of sexually transmitted infections (STI) in this region, is high. However, data are insufficient to validate this hypothesis.

The study of viral hepatitis prevalence along the Maroni River (MaHeVi) hence took place in a large strip of rural villages in an Amazonian context, with a complex and interesting network of transnational relations. This included the mobile migrant population, mainly of Brazilian origin [19], living in the semi-permanent artisanal and small goldmining areas in this strip.

The main objective was to estimate the prevalence of hepatitis B (HBV), hepatitis C (HCV), hepatitis D (HDV), and HIV in the adult population on both the FG and Surinamese side of the Maroni River. The secondary objectives of the study were to assess KAP-B about viral hepatitis, and to identify specific determinants for an HBV, HCV, HDV, or HIV infection.

## Materials and methods

### Study design and population

The MaHeVi prevalence study was population-based, cross-sectional, multicentric, non-interventional, and minimally invasive. Participants, aged 18 years or older, living along the Maroni River in FG or Suriname, upstream from Apatou

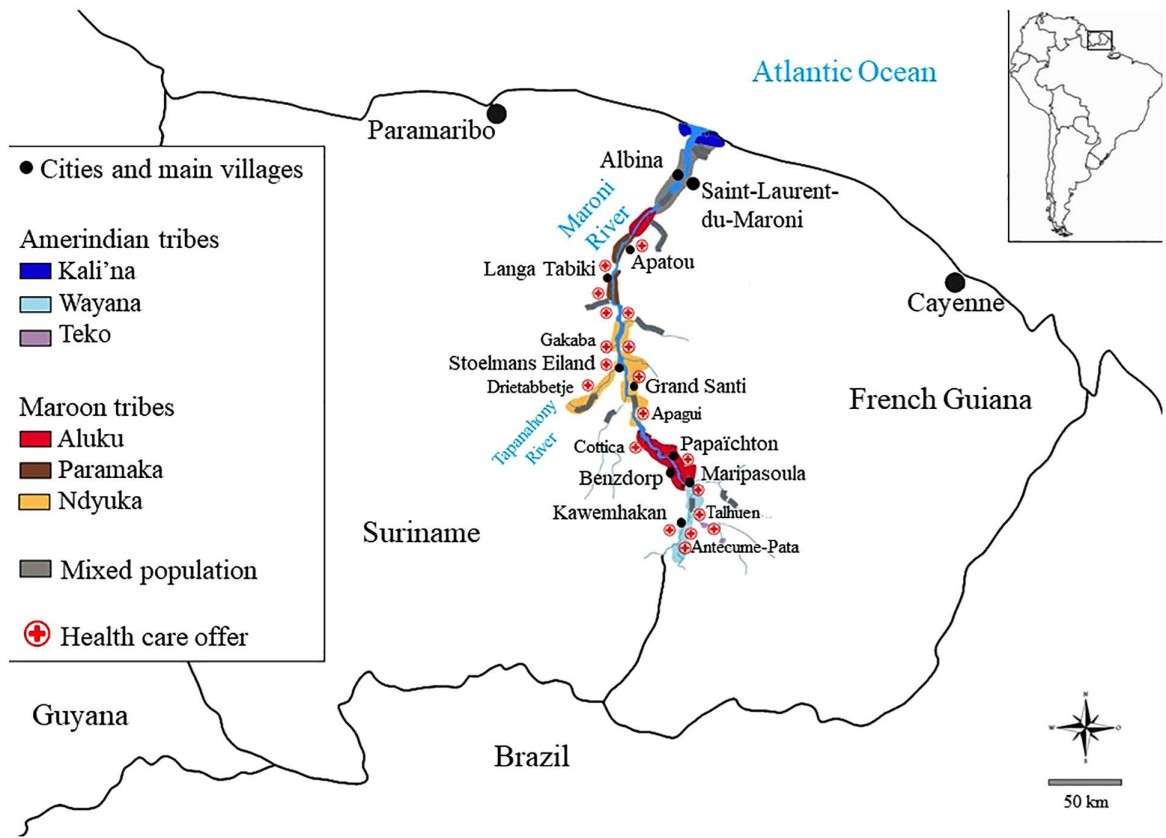

**Fig 1. Ethnic communities and healthcare posts along the Maroni River, border between Suriname and French Guiana.** Reprinted from [14] under a CC BY license, with permission from the original authors, original copyright 2012; modified by the authors of this work.

(Fig 1), planning to stay in the area for at least two months after inclusion, and providing informed consent, were recruited in French Guiana from the 16th January 2018–23rd November 2018 and in Suriname from 14th January 2018–8th February 2019, on a voluntary basis, following extensive preparation and a vast information campaign. No further exclusion criteria were applied beyond the inclusion criteria specified above. Carrying out a study in such a logistically and culturally challenging context called for appropriate planning and communication to ensure adequate information and gain the support of local populations in a context where mistrust of researchers is commonplace. We have described this preparation in detail elsewhere [20]. Participants completed a standardized structured questionnaire, available in local languages, which was developed based on information from focus groups and in-depth interviews done by anthropologists in the preliminary study [20]. Due to cultural and linguistic specificities and lower literacy levels in the interior, questionnaires were administered by trained cultural mediators. Subsequently, blood samples were collected for viral hepatitis screening and, optionally, HIV. We chose the opt-out option for HIV testing based on data indicating this may increase test acceptance rate [21].

### Testing methods and laboratory analyses

We used Dried Blood Spots (DBS) instead of conventional venous blood sampling for logistical and acceptability reasons [22]. Indeed, access to centrifugation facilities and refrigerated storage space was mostly non-existent, and timely sample transfer to a fully equipped laboratory was not guaranteed. Seven drops of capillary blood obtained by finger prick were collected by a trained nurse on a blotting paper Whatman 903™ Protein Saver Cards (Whatman GmbH, Dassel,

Germany). After drying, the DBS were stored at 4°C while in the field, transported in an icebox to Cayenne, stored at −20°C and sent on dry ice to Montpellier for laboratory analysis. HBV, HCV, HDV, and HIV serologies were performed, and in case of a positive result, viral load was measured.

All laboratory analyses were done at the laboratory of Virology, Montpellier University Hospital, France. For serologies, eight punches of 6 mm in diameter were eluted in 1000 μL of PBS and for molecular analyses, two punches of 6 mm in diameter were eluted in 240 μL of PBS-BSA (10%)/ Tween20 (0.05%) buffer solution. HBV and HCV extraction were performed in accordance with the manufacturer's instructions, with respectively QIAamp® DNA Mini Kit and QIAamp® RNA Mini Kit (QIAGEN, Hilden, Germany).

Serological testing for HBV, HCV, HIV and HDV was performed using commercial assays according to the manufacturers' instructions. Whenever applicable, manufacturer-recommended cutoffs were used. For some assays, these thresholds were adjusted for DBS eluates, as the kits were validated for serum samples and may be subject to DBS-related matrix effects. Specific cutoff values for each assay are provided in S2 Table.

**Hepatitis B virus serology and molecular analysis.** Samples were tested for hepatitis B using LIAISON® XL Murex HBsAg Quant kit (DiaSorin, Saluggia, Italy) for Hepatitis B Surface Antigen (HBsAg). For Hepatitis B surface antibodies (anti-HBs Ab) and anti-HBc antibodies (anti-HBc Ab), two assay kits were used for the same serology during the study period, following platform replacement: LIAISON® XL Murex Anti-HBs kit (DiaSorin, Italy), followed by Alinity i Anti-HBs Reagent Kit (Abbott, Chicago, IL, USA); and ADVIA Centaur® HBc Total kit (Siemens, Tarrytown, NY, USA) followed by Alinity i Anti-HBc II Reagent Kit (Abbott, USA), respectively. The detection limit of anti-HBs Ab on DBS samples was estimated at 50 IU/mL.

All HBsAg positive samples were further tested for hepatitis B e-antigen (HBe) using LIAISON® XL Murex HBeAg kit (DiaSorin, Italy) and HBV DNA detection and quantification using PUMA HBV real time PCR kit (Omunis, Clapiers, France).

Serologies were interpreted according to possible infection status: (i) acute/chronic HBV infection: positive HBsAg and positive anti-HBc Ab; (ii.) resolved hepatitis: negative HBsAg and positive anti-HBc Ab; (iii) HBV exposed: anti-HBc Ab positive regardless of HBsAg status, (iv) HBV vaccinated if anti-HBs positive and all other markers negative; (v) HBV susceptible if all markers are negative.

**Hepatitis C virus serology and molecular analysis.** Samples were tested for anti-HCV antibodies (anti-HCV Ab) using LIAISON® XL MUREX HCV Ab kit (DiaSorin, Italy). Positive samples were tested for HCV RNA detection and quantification using PUMA HCV real time RT-PCR kit (Omunis, Clapiers, France).

**Hepatitis D virus serology.** HBsAg positive samples were tested for anti-HDV antibodies (anti-HDV Ab) using ETI-AB-DELTAK-2 (EIA) anti-HDV assay (DiaSorin, Italy).

**HIV serology.** Samples were tested using HIV p24 antigen and HIV antibodies (HIV Ab) with fourth-generation combined EIA LIAISON® XL MUREX HIV Ab/Ag kit (DiaSorin, Italy) and if positive, confirmed by HIV-Ab positivity and determination of HIV-1, HIV-2, or HIV-1/2 infection with the Geenius™ HIV1/2 Confirmatory Assay (Bio-Rad, France).

## Determinants of HBV, HCV, HDV, and HIV infection, and KAP-B regarding HBV and HCV infection

Through the standardized interviewer-led questionnaire, we recorded self-reported social and demographic characteristics of participants, known or potential determinants associated with HBV, HCV, HDV, and HIV infection, e.g., unsafe sex practices: early sexual initiation, multiple sex partners during the last year, not using a condom during the last intercourse, transactional sex during the last intercourse, lifetime history of STIs or bleeding during intercourse; alcohol (Alcohol Use Disorders Identification Test-Concise (AUDIT-C test [23]) and (injecting) drug-use (IDU). Body modifications (tattoos and piercings, and whether they were performed in regulated or unregulated conditions); cultural practices, e.g., *koti-* a traditional medical practice involving scarification followed by application of medicine- and penile implants (*bugrus*), all performed with potentially unsterilized equipment, vaginal steam baths using astringent plants; previous blood transfusions

or dialysis; and sharing personal hygiene equipment such as toothbrushes, razors or nail clippers. were also evaluated as potential determinants of these infections. KAP-B regarding HBV and HCV infections were assessed using a standardized closed questionnaire, with "yes", "no", or "do not know/maybe" answers. Participants were assigned an ethnic group based on their mother tongue, as described previously in studies done on the Maroni River [15].

## Sample size

For HBV, to get a precision of 0.7% as expressed by the 95% CI in estimating an expected HBsAg seropositivity prevalence of 3%, we needed to include 2282 participants. For HCV, the sample size corresponding to 0.4% precision in estimating an expected HCV prevalence of 1%, is 2377. We therefore, aimed to include a total of 2500 persons along the Maroni River (approximately 12% of the targeted population), which enabled us to achieve sufficient precision in our prevalence estimates.

## Statistical analysis

Categorical variables were described as frequencies and percentages, and continuous variables as means with standard-deviations (SD), or medians with interquartile range (IQR) or range, depending on the distribution. We calculated the HBV, HCV, and HIV prevalence, and their corresponding 95% CI, corrected by post-stratification weighting to correct for sampling selection bias. Post-stratification weighting was carried out on the population structure in terms of age and sex of each geographical unit, based on 2012 census data from Suriname (for each resort) and 2018 census data for French Guiana (for each municipality).

We compared participant characteristics across HBV exposure groups (acute or chronic infection, or resolved infection), acute or chronic HCV infection, and HIV infection. Risk-factor analyses focused on HBV-exposed versus HBV-susceptible individuals, and HIV-positive versus HIV-negative individuals; determinants of HCV infection were not evaluated due to the very small number of HCV cases. Odds ratios (ORs) and 95% CIs were computed using bivariable logistic regression, and linearity of continuous variables was assessed. Variables with $p < 0.15$ in bivariable analyses were entered into the multivariable model. Those with >20% missing data or with counterintuitive associations (e.g., protective effects of piercings, or toothbrush sharing) were excluded. Collinearity among covariates was examined in both the full and final models. Variables were then manually removed using a backward procedure, starting with the least significant, until only those with $p < 0.05$ remained. Model fit was evaluated using the Hosmer–Lemeshow test. For HIV analyses, bivariable and multivariable logistic regressions were performed using the Firth correction for rare events [24]. A p-value<0.05 was considered statistically significant. Statistical analyses were conducted using SAS software Version 9.4 (SAS Institute Inc., Cary, NC, USA).

## Ethical and regulatory aspects

The MaHeVi study (ClinicalTrials.gov Identifier: NCT05002907) was approved by the Ethics Committee of the Ministry of Health of Suriname (VG 023–16) and by the Comité d'Evaluation Ethique de l'Institut National de la Santé et de la Recherche Médicale (IRB00003888). All participants provided written informed consent prior to their inclusion in the study. Results were delivered individually and confidentially to participants by the study doctors or the general practitioner at their respective health care center within 4–6 weeks after testing, allowing participants with positive results to be followed and treated according to standard clinical care of the participating country. All samples were centralized and stored by the CRB Amazonie, the biobank of Cayenne Hospital Center.

## Results

We included 2289 participants between January 2018 and February 2019. Three participants were excluded due to an empty or lost questionnaire hence a total of 2286 participants were analyzed. Six percent (n = 134) participants opted out

HIV-testing. There were significant differences in country of birth and residence, and educational level between participants who opted out for HIV testing compared to those that did not. There were no differences in sex, age, work, or other determinants. S1 Table shows inclusion numbers by site and by country.

## Study population

There were slightly more women included in the study (56.3%, n = 1285) than men, and the mean age (±SD) was 42.8 ± 14.4 years (Table 1). The majority of the participants (72.7%) spoke one of the Maroon languages, mainly Ndjuka, and 8.3% spoke one of the Amerindian languages, primarily Wayana. Almost 60% (n = 1319) had no formal education or only primary education. Table 2 describes known and potential determinants, including substance use, medical history, body modifications, and sexual behavior in the MaHeVi population, according to viral hepatitis and HIV infection.

## Prevalence of hepatitis B, C, D, and HIV, and HBV vaccination status

Crude DBS results are presented in S2 Table. Overall, 2.01% (95%CI: 1,48–2.67; n = 46) participants tested positive for HBsAg, and 2.61% and 1.56% for men and women respectively (p = 0.08). After post-stratification weighting, the adjusted prevalence estimate for HBsAg seropositivity was 2.08% (95%CI: 1.49–2.66). Almost half of HBsAg seropositive participants (n = 20) had a detectable HBV VL; (median: 8.7909 IU/mL; IQR: 147−3,03.10$^7$), and five had a positive HBeAg. None of the HBsAg-positive participants had an HDV coinfection.

More than a quarter of the participants (28.3%, n = 646; adjusted prevalence 25.06% (95%CI: 20.89–29.24) had anti-HBc Ab who tested HBsAg seronegative, of which 51.6% (n = 333) had detectable anti-HBs Ab). A total of 23.5% (n = 538) had serological evidence of HBV vaccination, 43.1% (n = 985) were susceptible to HBV infection, having no HBV serological markers, 0.9% (n = 21) had no markers of infection but an unknown anti-HBs Ab serology, and 2.2% (n = 50) had an HBV serology that was difficult to interpret (S3 Table). Excluding participants who had a non-interpretable HBV status, 27.14% (95%CI: 22.74–31.54; n = 692) are or were infected with HBV, i.e., HBV exposed.

Five participants had anti-HCV Ab (0.22%), corresponding to an adjusted HCV seroprevalence of 0.13% (95%CI: 0.00–0.27), of which four had a detectable HCV viral load (range: 52.100–422.000 IU/mL); one woman and four men tested anti-HCV positive. Two anti-HCV positive participants were of Maroon descent, two of French descent and one of other descent.

Most participants (94%; n = 2152/2286) opted-in for HIV testing; 13/2152 had anti-HIV Ab, of which 12 were confirmed with the Geenius Confirmatory Assay, corresponding to an adjusted prevalence of 0.65% (95%CI: 0.00–1.40) overall, and a 0.54% prevalence among 18–49 years old. HIV prevalence was 0.64% in males and 0.50% in females (p = 0.30), and 1.36%, 0.45% and 0% in participants of Brazilian, Maroon, and Indigenous descent, respectively, and 1.05% in participants from other descent. There were no co-infections.

23.5% (n = 538) of participants had DBS serological markers indicating an effective vaccination (S3 Table). Vaccinated participants were significantly younger (39 years old, IQR [30–50]) compared to susceptible participants and participants with a current or resolved HBV infection (43 years old [32–53]; p < 0.001). Participants from Amerindian descent were more frequently vaccinated (57.2%; n = 95), compared to Maroon (18.0%, n = 291) or other participants (34.9%, n = 149; p < 0.0001). Participants with a middle school education level or higher, were more likely to be vaccinated (30.3%, n = 275), compared to participants who did not go beyond primary school (19.7%; n = 251; p < 0.001).

## Determinants of HBV exposure, HCV infection, and HIV infection

In multivariable analysis, age, sex, ethnicity, and educational level were significantly associated with HBV exposure, compared to those who were HBV-susceptible (Table 3). Older age was significantly associated with HBV exposure (per 10-year increase, aOR:1.27, 95%CI: 1.18–1.37, p < 0.0001). Males had significantly higher odds of HBV exposure

**Table 1. Socio-demographic characteristics of the MaHeVi study population along the Maroni river.**

| Socio-demographic characteristics | | Study population | HBV infection (HBsAg +) | Resolved HBV infection | HCV infection | HIV infection[&] |
|---|---|---|---|---|---|---|
| n (%) or mean±SD (range) | | (n=2286) | (n=46) | (n=646) | (n=5) | (n=12) |
| **Sex** | Male | 998 (43.7) | 26 (56.5) | 297 (46.0) | 4 (80.0) | 6 (50.0) |
| | Female | 1285 (56.3) | 20 (43.5) | 349 (54.0) | 1 (20.0) | 6 (50.0) |
| **Age in years** | | 42.8±14.4 (18-95) | 40.9±13.1 (23-88) | 47.5±15.0 (18-92) | 59.0±7.8 (53-68) | 52.3±13.0 (35-79) |
| **Mother tongue (Ethnicity)** | Maroon* | 1655 (72.7) | 36 (78.3) | 513 (79.5) | 2 (40.0) | 7 (58.3) |
| | Amerindian† | 189 (8.3) | 1 (2.2) | 17 (2.6) | 0 (0) | 0 (0) |
| | French | 45 (2.0) | 0 (0) | 3 (0.5) | 2 (40.0) | 0 (0) |
| | Dutch | 27 (1.2) | 1 (2.2) | 10 (1.6) | 0 (0) | 1 (8.3) |
| | Portuguese | 231 (10.1) | 6 (13.0) | 65 (10.1) | 0 (0) | 3 (25.0) |
| | Other | 132 (5.8) | 2 (4.4) | 37 (5.7) | 1 (20.0) | 1 (8.3) |
| **Country of birth** | French Guiana | 766 (33.6) | 16 (34.8) | 154 (23.9) | 2 (40.0) | 4 (36.4) |
| | Suriname | 1175 (51.5) | 22 (47.8) | 403 (62.5) | 2 (40.0) | 4 (36.4) |
| | Brazil | 245 (10.7) | 6 (13.0) | 68 (10.5) | 0 (0) | 3 (27.3) |
| | Other | 94 (4.1) | 2 (4.4) | 20 (3.1) | 1 (20.0) | 0 (0) |
| **Country of residence** | French Guiana | 1422 (62.4) | 27 (58.7) | 350 (54.5) | 3 (60.0) | 6 (50.0) |
| | Suriname | 841 (36.9) | 19 (41.3) | 291 (45.3) | 2 (40.0) | 6 (50.0) |
| | Other | 15 (0.7) | 0 (0) | 1 (0.2) | 0 (0) | 0 (0) |
| **School level** | Never been to school | 797 (35.4) | 16 (34.8) | 272 (42.6) | 1 (25.0) | 5 (41.7) |
| | Primary | 522 (23.2) | 13 (28.3) | 166 (26.0) | 1 (25.0) | 3 (25.0) |
| | Middle school | 484 (21.5) | 8 (17.4) | 127 (19.9) | 1 (25.0) | 2 (16.7) |
| | High school | 356 (15.8) | 6 (13.0) | 62 (9.7) | 1 (25.0) | 2 (16.7) |
| | Higher education | 91 (4.0) | 3 (6.5) | 11 (1.7) | 1 (25.0 | 0 (0) |
| **Professional activity** | Don't work | 985 (43.5) | 16 (35.6) | 285 (44.3) | 0 (0) | 5 (41.7) |
| | Work | 1257 (55.5) | 29 (64.4) | 354 (55.1) | 5 (100) | 7 (58.3) |
| | Student/Training | 23 (1.0) | 0 (0) | 4 (0.6) | 0 (0) | 0 (0) |

HBsAg: hepatitis B surface antigen, HCV: hepatitis C virus, HIV: human immunodeficiency virus, SD: Standard Deviation; *Ndjuka, Aluku, Pamaka, Saramaka; †Wayana, Teko, Trio, Apalai; & Denominator: 2152 participants opted in for HIV testing. Percentages are calculated excluding missing data – percentages add up to 100%.

compared to females (aOR:1.34, 95%CI: 1.10–1.65, p=0.006), and those of Maroon or other ethnicity had higher odds of HBV exposure compared to those of Amerindian descent – aOR:2.22 (95%CI: 1.27–4.06) and aOR:2.94 (95%CI: 1.61–5.59) respectively (p=0.002). Participants with a higher education level had significantly lower odds of HBV exposure compared to those with a lower education level (aOR:0.71, 95%CI: 0.56–0.90, p=0.005). Although being in a polygamous relationship was significantly associated with HBV exposure in bivariable analysis, due to more than 20% missing answers, this determinant was not included in the multivariable model. Alcohol and drug use, nosocomial determinants – blood transfusion, dialysis, and dental care in non-regulated setting, potential sexual risk factors – early sexual initiation, condomless sex, transactional sex, history of STI (lifetime), same-sex partner, bleeding during intercourse and having *bugrus*, and traditional customs – vaginal steam baths and scarification – tattoos, piercings, and sharing of toothbrushes, nail clippers or razors/epilators were not associated with increased odds of HBV exposure. Subgroup analyses stratified by sex (S4 Table) and ethnic group (S5 Table) revealed broadly similar associations, although the magnitude and significance of these factors differed between subgroups.

**Table 2. Hepatitis B, C and HIV determinants in the MaHeVi study population along the Maroni River.**

| HBV, HCV, and HIV known or suspected determinants | | Study population | HBV infection (HBsAg +) | Resolved HBV infection | HCV infection | HIV infection |
|---|---|---|---|---|---|---|
| n (%) or median [IQR] | | (n = 2286) | (n = 46) | (n = 646) | (n = 5) | (n = 12) |
| **Alcohol and drugs** | **Alcohol use disorders (positive AUDIT-C# test)** | **578 (27.6)** | **14 (34.2)** | **128 (21.3)** | **2 (40.0)** | **2 (16.7)** |
| | **Drug use during the past month** | **199 (8.8)** | **5 (11.1)** | **59 (9.2)** | **3 (60.0)** | **1 (8.3)** |
| | marijuana | 159 (7.0) | 5 (11.1) | 41 (6.4) | 2 (40.0) | 1 (8.3) |
| | crack | 22 (1.0) | 0 (0) | 11 (1.7) | 0 (0) | 0 (0) |
| | cocaine | 5 (0.2) | 0 (0) | 4 (0.6) | 0 (0) | 0 (0) |
| | other drug | 36 (1.6) | 1 (2.2) | 14 (2.2) | 1 (20.0) | 0 (0) |
| | **Injection drug use (ever)** | **15 (0.7)** | **0 (0)** | **3 (0.5)** | **2 (50.0)** | **0 (0)** |
| **Health care and nosocomial risks** | **Blood transfusion&** | **280 (12.4)** | **5 (11.1)** | **81 (12.7)** | **1 (20.0)** | **2 (18.2)** |
| | before 1990 | 37 (1.7) | 0 (0) | 15 (2.4) | 1 (20.0) | 1 (9.1) |
| | 1990-2000 | 47 (2.1) | 0 (0) | 19 (3.0) | 0 (0) | 1 (9.1) |
| | >2000 | 168 (7.6) | 4 (9.1) | 35 (5.5) | 0 (0) | 0 (0) |
| | unknown year | 15 (0.7) | 1 (2.3) | 8 (1.3) | 0 (0) | 0 (0) |
| | **Dialysis** | **54 (2.4)** | **0 (0)** | **16 (2.5)** | **0 (0)** | **0 (0)** |
| | **Dental care outside of official health care facility** | **277 (12.3)** | **2 (4.4)** | **85 (13.3)** | **3 (60.0)** | **1 (8.3)** |
| | at home/with a friend | 38 (1.7) | 0 (0) | 13 (2.1) | 0 (0) | 0 (0) |
| | in a shop/traveling practitioner | 183 (8.2) | 2 (4.4) | 58 (9.2) | 3 (60.0) | 1 (8.3) |
| **Sexual risk factors** | **Age at first sexual encounter** | **16 [14-18]** | **16 [15-18]** | **16 [14-18]** | **17 [14-19]** | **16 [13-17]** |
| | **In a polygamous relationship** | **306 (21.6)** | **6 (22.2)** | **957 (24.7)** | **0 (0)** | **1 (14.3)** |
| | **Several partners in last year** | **414 (23.1)** | **10 (27.0)** | **112 (22.5)** | **2 (50.0)** | **1 (11.1)** |
| | 2-3 partners | 212 (12.3) | 5 (14.3) | 56 (11.6) | 0 (0) | 1 (11.1) |
| | ≥4 partners | 138 (8.0) | 3 (8.6) | 40 (8.3) | 2 (50.0) | 0 (0) |
| | **Casual partner (last intercourse)** | **387 (19.1)** | **8 (19.5)** | **100 (17.6)** | **3 (75.0)** | **2 (18.2)** |
| | **No condom use (last intercourse)** | **1456 (70.2)** | **30 (69.8)** | **411 (71.1)** | **2 (40.0)** | **7 (63.6)** |
| | **Transactional sex** (last intercourse)** | **139 (7.3)** | **6 (15.0)** | **41 (7.7)** | **0 (0)** | **2 (22.2)** |
| | **History of STI (lifetime)** | **310 (14.3)** | **8 (18.6)** | **86 (13.9)** | **4 (80.0)** | **2 (16.7)** |
| | **Sex with same-sex partner (lifetime)** | **36 (1.6)** | **1 (2.3)** | **8 (1.3)** | **0 (0)** | **0 (0)** |
| | **Bleeding during intercourse^ (lifetime)** | **109 (11.3)** | **3 (23.1)** | **24 (10.3)** | **0 (0)** | **1 (20.0)** |
| | **Had/have *bugrus* (men) or partner with *bugrus*** | **234 (13.5)** | **7 (20.0)** | **66 (15.5)** | **0 (0)** | **0 (0)** |
| **Customs, rituals, hygiene, and cosmetic risks** | **Vaginal steam baths^** | **974 (80.5)** | **13 (81.3)** | **292 (87.7)** | **1 (100)** | **4 (66.7)** |
| | everyday | 707 (59.7) | 11 (73.3) | 215 (66.0) | 1 (100) | 1 (16.7) |
| | regularly/after periods | 218 (18.4) | 1 (6.7) | 63 (19.3) | 0 (0) | 3 (50.0) |
| | after childbirth only | 24 (2.0) | 0 (0) | 7 (2.2) | 0 (0) | 0 (0) |
| | **Scarification (*koti*)** | **166 (7.7)** | **1 (2.3)** | **55 (9.1)** | **1 (20.0)** | **0 (0)** |
| | **Tattoos** | **535 (23.6)** | **10 (21.7)** | **128 (20.0)** | **3 (60.0)** | **0 (0)** |
| | At least 1 done in an unregulated setting | 358 (15.8) | 7 (15.2) | 89 (13.9) | 2 (40.0) | . |
| | All done in regulated setting | 177 (7.8) | 3 (6.5) | 39 (6.1) | 1 (20.0) | . |
| | **Piercings** | **1504 (66.3)** | **27 (58.7)** | **401 (62.9)** | **3 (60.0)** | **5 (41.7)** |
| | At least 1 done in a non-regulated setting | 1283 (56.5) | 20 (43.5) | 346 (54.2) | 3 (60.0) | 5 (41.7) |
| | All done in regulated setting | 221 (9.7) | 7 (15.2) | 55 (8.6) | 0 (0) | 0 (0) |

*(Continued)*

**Table 2.** (Continued)

| HBV, HCV, and HIV known or suspected determinants | | Study population | HBV infection (HBsAg +) | Resolved HBV infection | HCV infection | HIV infection |
|---|---|---|---|---|---|---|
| | Sharing of toothbrushes | 214 (9.4) | 3 (6.5) | 34 (5.3) | 0 (0) | 0 (0) |
| | Sharing of nail clippers | 878 (39.0) | 18 (39.1) | 223 (35.2) | 2 (40.0) | 2 (16.7) |
| | Sharing of razor/epilator | 200 (8.9) | 5 (11.1) | 44 (7.0) | 1 (20.0) | 0 (0) |

HBsAg: hepatitis B surface antigen, HCV: hepatitis C virus, HIV: human immunodeficiency virus, IQR: Inter quartile range, STI: sexually transmitted infections; #Positive AUDIT-C test: score ≥3 for women and ≥4 for men; **Transactional sex was defined as having paid for or having been paid in exchange for sex; ^Female only determinant; &In Suriname, blood products are screened for HBV and HIV since 1988 and for HCV since 1998; in French Guiana, blood products are screened for HBV, HIV, and HCV since 1971, 1985, and 1990 respectively; Percentages are calculated excluding missing data – and add up to 100%.

In multivariable analysis, older age was significantly associated with HIV infection (per 10-year increase aOR:1.64, 95%CI:1.11–2.44), and participants of other ethnic groups (aOR:3.81, 95%: CI 1.14–12.26); compared to Amerindian and Maroon ethnic groups had significantly higher odds of having an HIV infection (p = 0.020; Table 4). No other determinants were associated with an HIV infection.

With only five participants infected with HCV, we did not perform statistical analysis to assess risk factors for HCV infection in this population. However, HCV-infected participants were mostly male (n = 4 or 80%) and older (59.0 ± 7.8 years old) than our global study population (43.7% male and 42.8 ± 14.4 years old). Furthermore, we noted an overrepresentation of drug use (n = 3 or 60%), injection drug use (n = 2 or 50%), and dental care outside of official care facility (n = 3 or 60%) among our HCV-positive participants compared to the global population study (8.8%, 0.7%, and 12.3% respectively).

## KAP-B regarding viral hepatitis

Knowledge and awareness about viral hepatitis were very low (S6 Table); approximately one in three participants had heard about HBV and one in five participants about HCV. Of those who had heard about it, one-third believed mosquito bites to play a role in transmission. Approximately half correctly cited mother-to-child transmission and IDU as risk factors for viral hepatitis, and over half correctly identified unprotected sex as a risk factor. Self-reported testing rates were very low: 11.8% for HBV and 8.3% for HCV. Among 2116 responders, 255 (12.1%) said they were vaccinated against HBV.

## Discussion

This study among adults living along the remote and rural French Guiana-Suriname border shows an adjusted HBsAg, resolved HBV infection, HCV, and HIV seroprevalence of 2.08% (95%CI: 1.49–2.66), 25.06% (95%CI: 20.89–29.24), 0.13% (95%CI: 0.00–0.27) and 0.65% (95%CI: 0.0–1.40) respectively. There were no HDV infections. Prevalence of all HBV, HCV and HIV were comparable, although somewhat lower, to the prevalence found in the urban areas of FG and Suriname; in the urban areas, HBsAg was 3% and 3.2% respectively in FG and Suriname [11], anti-HBc was 33% in pregnant women in Saint-Laurent-du-Maroni in FG [25] and 24.5% (95%CI: 22.7–26.4%) in Suriname [11], anti-HCV seropositivity was 0.67% in Cayenne [26] and 1.0% (95%CI: 0.6–1.58) in the (Emergency Department) ED population in Paramaribo, Suriname [12], however, prevalence was highest in people of Javanese descent, of which there were only fifteen in this study. In fact, in the ED study, in Afro-Surinamese participants, HCV prevalence was an estimated 0.69% (95%CI: 0.17–2.91). HIV prevalence in the urban areas was 1.18%−1.35% in FG [2] and 1.6% (95%CI: 1.3–1.9) HIV seropositivity overall [3].

**Table 3. Logistic regression models for risk factors of HBV exposure along the Maroni.**

| Known or suspected risk factors | | HBV-susceptible | HBV-exposed | Bivariable models | Multivariable model |
|---|---|---|---|---|---|
| (N analyzed in bivariable analyses) | | N (%) or median [IQR] | | OR [95%CI], p-value | aOR [95%CI], p-value |
| For HBV exposure (n = 1677) | | (n = 985) | (n = 692) | | (n = 1599) |
| Age (n = 1626) | (per 10-year increase) | 40 [30-50] | 46 [36-57] | **1.32 [1.23–1.41], p < 0.0001** | **1.27 [1.18–1.37], p < 0.0001** |
| Sex | Male | 405 (41.2) | 323 (46.7) | **1.25 [1.03–1.52], p = 0.027** | **1.34 [1.10–1.65], p = 0.006** |
| (n = 1674) | Female (RG) | 577 (58.8) | 369 (53.3) | – | |
| Mother tongue | Maroon | 775 (78.9) | 549 (79.5) | **2.09 [1.21–3.60], p = 0.008** | **2.22 [1.27–4.06], p = 0.007** |
| (n = 1673) | Other | 154 (15.7) | 124 (18.0) | **2.37 [1.32–4.25], p = 0.004** | **2.94 [1.61–5.59], p = 0.0007** |
| | Amerindian (RG) | 53 (5.4) | 18 (2.6) | **- (global p = 0.015)** | **- (global p = 0.002)** |
| Education level | Middle/High/University | 416 (42.9) | 217 (31.7) | **0.62 [0.50–0.76], p < 0.0001** | **0.71 [0.56–0.90], p = 0.005** |
| (n = 1654) | No education/Primary (RG) | 554 (57.1) | 467 (68.3) | – | – |
| In a polygamous relationship | Yes | 118 (19.4) | 103 (24.5) | **1.35 [1.00–1.82], p = 0.049** | >20% missing data |
| (n = 1029) | No (RG) | 491 (80.6) | 317 (75.5) | – | – |
| Piercing | At least 1 done in unregulated setting | 378 (38.6) | 318 (46.5) | **0.72 [0.59–0.88], p = 0.013** | not relevant |
| (n = 1664) | None or done in regulated setting (RG) | 602 (61.4) | 366 (53.5) | – | – |
| Sharing toothbrush | Yes (often or sometimes) | 77 (7.9) | 37 (5.4) | **0.67 [0.44–1.00], p = 0.049** | not relevant |
| (n = 1664) | No (RG) | 900 (92.1) | 650 (94.6) | | – |
| Alcohol use disorder | Yes | 232 (25.7) | 142 (22.1) | 0.82 [0.65–1.04], p = 0.109 | – |
| (n = 1546) | No (RG) | 672 (74.3) | 500 (77.9) | – | – |
| At risk transfusion (before 1990) | Yes | 11 (1.2) | 15 (2.2) | 1.97 [0.90–4.30], p = 0.092 | – |
| (n = 1632) | No (RG) | 948 (98.8) | 658 (97.8) | – | – |
| Transactional sex (last intercourse) | Yes | 46 (5.7) | 47 (8.2) | 1.48 [0.97–2.25], p = 0.070 | – |
| (n = 1384) | No (RG) | 763 (94.3) | 528 (91.8) | – | – |
| Age at first intercourse < 15 YO | Yes | 272 (32.3) | 164 (28.6) | 0.84 [0.67–1.06], p = 0.145 | – |
| (n = 1416) | No (RG) | 571 (67.7) | 409 (71.4) | – | – |

CI: confidence interval, HBV: hepatitis B virus, IQR: inter quartile range, OR: odds ratio;RG: reference group; P-values in bold are statistically significant; All selected known or potential risk factors with a p-value < 0.15 are presented in this table; *Ndjukatongo, Aluku, Pamaka, Saramaka; †Wayana, Teko, Trio, Apalai.

Older age, male sex, Maroon or other ethnicity (compared to Amerindians) and lower education level were all significantly associated with HBV exposure in this transborder population. Interestingly, apart from male sex, the same determinants: older age, Maroon ethnicity and lower education were found to be associated with HBV exposure in the ED population in the urban region Paramaribo, suggesting similar HBV prevalence and modes of transmission throughout Suriname. Similarly, HBV was also more often seen in males in the urban areas of French Guiana, and within a subgroup of pregnant women, HBV was more often seen in Asians, Haitians and Maroons [27].

 

**Table 4. Logistic regression models for risk factors HIV infection along the Maroni.**

| Known or suspected risk factors | | HIV-positive | HIV-negative | Bivariable models | Multivariable model |
|---|---|---|---|---|---|
| (N analyzed in bivariable analyses) | | N (%) or median [IQR] | | OR [95%CI], p-value | aOR [95%CI], p-value |
| **For HIV Infection (n=2150)** | | **(n=12)** | **(n=2138)** | | **(n=2087)** |
| Age (n=2095) | (per 10-year increase) | 48 [44-61] | 42 [32-52] | **1.52 [1.06–2.18], p=0.019** | **1.64 [1.11–2.44], p=0.01** |
| Sex | Male | 6 (50.0) | 936 (43.8) | 1.28 [0.43–3.81], p=0.657 | 1.09 [0.35–3.36], p=0.881 |
| (n=2147) | Female (RG) | 6 (50.0) | 1199 (56.2) | – | – |
| Mother tongue* | Portuguese and other | 5 (41.7) | 404 (19.0) | **3.13 [1.04–9.48], p=0.043** | **3.81 [1.14–12.26], p=0.020** |
| (n=2144) | Amerindian† or Maroon‡ | 7 (58.3) | 1728 (81.0) | – | – |
| Tattoos | Yes | 0 (0.0) | 516 (24.3) | 0.12 [0.01–2.11], p=0.149 | – |
| (n=2133) | No (RG) | 12 (100) | 1605 (75.7) | – | |
| Piercing | Yes | 5 (41.7) | 726 (34.2) | 0.38 [0.13–1.15], p=0.087 | – |
| (n=2135) | No (RG) | 7 (58.3) | 1397 (65.8) | – | |
| Transactional sex (LI) | Yes | 2 (22.2) | 131 (7.4) | 4.17 [0.98–17.73], p=0.053 | – |
| (n=1786) | No (RG) | 7 (77.8) | 1646 (92.6) | – | |
| Age at first intercourse <14 YO | Yes | 4 (44.4) | 321 (17.5) | 3.86 [1.10–13.53], p=0.034 | >20% missing data |
| (n=1416) | No (RG) | 5 (55.6) | 1518 (82.5) | – | – |

CI: confidence interval, HBV: hepatitis B virus, HIV: human immunodeficiency virus, IQR: inter quartile range, OR: odds ratio, LI: last intercourse; P-values in bold are statistically significant; Age and sex were forced into the final multivariable model; *Mother tongue groups with very small sample sizes or no HIV cases were combined to avoid sparse-data issues; †Wayana, Teko, Trio, Apalai; ‡Ndjukatongo, Aluku, Pamaka, Saramaka;.

The trend between older age and HBV exposure has also been observed in other Latin American countries [28]. This association is likely due to cumulative exposure to HBV risk factors over time. Furthermore, the relatively high rate of resolved HBV infection identified in this study suggests that the majority of participants were likely infected later in life. Specifically, the ratio of HBsAg prevalence (2.08%) to resolved HBV infection (25.06%) aligns with the expected 5–10% chronic HBV persistence rate following adult-acquired infections, supporting the interpretation that HBV acquisition predominantly occurred during adulthood rather than through early-life transmission.

Males had significantly higher HBV exposure compared to females. This has been seen in other countries, including neighboring Brazil [28], with increased sexual risk behavior identified as possible determinant. In our study, polygamy was a significant determinant in bivariable analysis; however due to the large proportion of missing data, it could not be included in the multivariable model. However, in the male subgroup analysis, this determinant was only borderline significant. Interestingly, in our study, none of the other potential sexual risk factors, including self-reported STI's, condomless sex, were associated with HBV or HIV. This finding should be interpreted with caution, as self-reporting of STIs is notoriously low, as the majority of people who have had an STI are in fact asymptomatic [29] and STI-associated stigma may have led to underreporting of STIs. Likewise, use of condom at last sexual encounter might also have been affected by this same social desirability bias [30], leading to a potential underestimation of this risk. These findings highlight the importance of tailoring awareness and testing strategies to include men.

Multiple sex partners or polygamy might also explain why Maroon and other ethnic groups demonstrated a significantly higher HBV exposure proportion compared to Amerindians. A previous KAP-B study along the Maroni also showed significant differences between the Maroon and Amerindian population regarding multiple sex partners [15]. Moreover, polygamy within the Maroon ethnic group is culturally accepted [31]. Elsewhere, ethnic disparities regarding HBV, and health care in general, are also well documented. In the US, for example, HBV prevalence among non-Hispanic blacks is significantly higher than other ethnic groups. Moreover, African-Americans and Mexican-Americans had significantly lower

vaccine coverage [32]. Similarly, in Suriname, afro-Surinamese are disproportionately affected by HIV compared to other ethnic groups. Approximately 70% of HIV infections occur among Afro-Surinamese descent [33], although they comprise only 38% of the population. Again, emphasizing safe sex practices is therefore crucial to reducing transmission of both HIV and HBV, as well as addressing these disparities. Interestingly, Amerindian participants in our study were more often vaccinated, which of course resulted in less HBV infection, and might be the primary reason for the disparity in HBV prevalence among the different ethnic groups. However, Maroon or other ethnic groups have higher odds of HBV exposure, underscoring the need for targeted HBV awareness campaigns, and also assessing whether catch-up vaccinations are warranted.

Remaining on the topic of awareness, there was a notable association between HBV exposure and lower educational level, highlighting the importance of targeted educational programs. This finding is consistent with studies conducted in comparable low- and middle- income settings [34]. Increasing knowledge and awareness of viral hepatitis is recommended as only one-third of the participants in our study had heard about HBV and one-fifth about HCV, with overall poor to average knowledge regarding transmission routes. The latter is in contrast with a previous KAP-B study on HIV among Maroni boatmen, where over 90% correctly identified unprotected sex as a transmission risk factor for HIV, reflecting the effect of the numerous preventive efforts on HIV in FG and Suriname [15,35,36]. Unfortunately, at the time, these initiatives focused exclusively on HIV and did not address viral hepatitis, or other STIs. Inclusive and tailored educational initiatives should be implemented to improve awareness in all populations, including guidance on safe sex practices and prevention of HBV as well as other sexually transmitted or blood-borne infections.

Given the low HCV prevalence, statistical analyses were not possible; however, notable determinants among anti-HCV seropositive participants included male sex, and older age – two determinants also seen in the ED population in Paramaribo [12] –, as well as injection drug use, and dental care in non-regulated settings. These factors warrant further investigation, and HCV screening should be considered for individuals exposed to such risks.

Regarding HIV, only older age and non-autochtonous ethnic groups were significantly associated with infection. This pattern may be explained by the delayed but rapid rise of HIV prevalence in pregnant women in the 1990s on the Maroni river [37], after which numerous HIV awareness and testing programs were introduced throughout the region. Presumably, these programs primarily targeted the local tribal and Amerindian population in this hard-to-reach area. At the same time, access to testing and treatment in these remote regions are more limited for hard-to-serve populations, particularly the non-autochthonous groups who are more often undocumented [38]. Both these disparities might explain the higher prevalence among non-autochthonous populations. To achieve elimination of both viral hepatitis and HIV, it is essential that all populations, including marginalized and undocumented groups, are reached through tailored programs ensuring equitable access to testing, prevention, and treatment services. Hopefully with increased awareness and knowledge of HIV and its risk factors, access to PrEP and condoms, prevalence of HIV will decrease [39].

We found no associations between hypothesized cultural risk factors – *bugrus,* vaginal steam baths, therapeutic *koti* scarifications-, tattoos, or piercings, and viral hepatitis or HIV infection in this study. While *bugrus* were expected to increase bloodborne infection risk in men during insertion [40], and STI risk in women, through mechanisms such as mucosal micro-abrasions and higher rates of condom rupture [41], we did not observe any significant association. However, there were a significant number of missing responses regarding *bugrus,* especially among female participants. Another hypothesized cultural practice – vaginal steam baths which were very often practiced – was not associated with the viral infections tested. Interestingly, despite the sensitive question, 94% of women responded to this question. Similarly, *kotis* were not associated with HBV exposure, although this association was seen in East Africa [42]. Contrary to expectations, tattoos and piercings – regardless of whether they were performed in regulated or non-regulated settings – were also not associated with any of the infections examined.

Overall, despite the presence of numerous transmission risk factors, these data are in alignment with the general assumptions that with less dense sexual networks, HIV or HBV prevalence in rural areas generally tend to be lower than

in urban areas [43]. A notable exception is the somewhat higher, although non-significant, HBsAg and HIV prevalence we found in the Brazilian population (mostly gold miners or *garimpeiros*), presumably reflecting the generally greater risk for STIs [44,45] in this highly precarious and mobile population where turnover is fast, transactional sex is frequent [45] and access to health care is challenging [7].

The non-existent HDV seropositivity we observed is noteworthy given the relatively high prevalence found in other studies conducted in South America. To our knowledge, this is the first hepatitis D seroprevalence study in Suriname. In FG, we are aware of one study on HDV infections, which showed no HDV coinfections in 19 HBsAg-seropositive patients out of 416 gold-miners from Brazil living along the FG Suriname border [44]. In South America, anti-HDV seropositivity has shown to vary significantly between and within areas, with an estimated pooled HDV prevalence of 22.4% among all hepatitis B carriers, 22.2% in Native indigenous communities and 32.1% in the Amazon basin, even though the most recent studies indicate a lower seroprevalence around 7% [46]. Further testing for HDV in the more urban areas in FG and Suriname would be of interest to verify whether our finding is only confined to the rural areas.

Another interesting finding in this study, was the low HBV vaccination coverage, although significantly higher than the 6.0% found in a previous study in Suriname in 2012 [11]. As only 23.5% had effective protection due to vaccination in our study, efforts to increase the vaccination rate are an important message for the health authorities, specifically for people born before 2005, since universal hepatitis B vaccination program was put into place for all newborns in Suriname in 2005 and in FG since 1994 [47]. Nonetheless, vaccination rates in Suriname's hard-to-reach interior are far lower than the urban areas [48]. In FG, children's HBV vaccination coverage is increasing but is still low in adults, especially on the Maroni [4]. This lower vaccination coverage level along the Maroni is not fully understood, but thought to be mainly caused by logistical constraints of these remote territories as well as vaccine hesitancy due to misinformation in FG [49] and Suriname [48].

The originality and strength of our study is that, whereas most published data on viral hepatitis and HIV in the region concern urban areas, our study focused on a rural transborder area. By using DBS for laboratory analysis, we ensured broad inclusion and testing of participants in even the very remote areas. Nevertheless, there are several limitations. First, although we called on cultural mediators to ensure participant comprehension and participation, refusals to answer certain questions were common. In addition, self-reported data may be affected by social desirability bias, which could have influenced the accuracy of participants' responses. Second, random sampling was not feasible in the context of hundreds of scattered small informal villages in the Amazon forest. In fact, the sampling was expanded per special request of the community leaders. This non-random sampling, may have introduced selection bias, potentially favoring the inclusion of individuals who are more integrated into the community, thereby limiting the participation for more marginalized persons. We thus attempted to correct prevalence estimates through sampling weights, however, this cannot fully remove the risk of bias. Third, some people living with HIV, HBV, or HCV, and aware of their infection, potentially did not participate in the study, which would have led to an underestimation of the true prevalence. Indeed, 6% opted out for HIV-testing and some of them did so declaring they were HIV-positive. However, the opt-out policy presumably led to a high participation rate; the HBV and HCV prevalence is however, likely more accurate than for HIV. Despite the opt-out, we believe that due to the large sample size, the HIV prevalence estimates are robust. Fourth, although the use of DBS was a strength and allowed testing of more than 2,000 participants, it also introduced methodological constraints. The serological assays were originally validated for serum rather than DBS eluates, and despite adjusting cutoffs to account for potential matrix effects, residual differences in analyte recovery and assay performance may still have resulted in some degree of mis-classification. Despite this, recent meta-analyses highlighted the exceptional diagnostic accuracy of HBsAg and HCV serology, as well as molecular analyses conducted using DBS, reporting over 98% sensitivity and specificity [50,51]. Fifth, as low level HBV DNA is difficult to detect using DBS, we were unable to collect data on occult HBV infection – defined as presence of HBV DNA in the liver with detectable or undetectable HBV DNA levels in serum, in individuals who are HBsAg seronegative [52]. In terms of public health, and prevention of HBV transmission, this aspect warrants further research. Lastly, HIV results should be interpreted with caution, due to the wide confidence intervals observed, which reflect the

small number of positive cases. The number of positives for HBV, HCV, and HIV were small and our analyses on determinants may therefore have suffered from low statistical power, and more data are needed.

## Conclusion

This is the first, large scale viral hepatitis and HIV prevalence study in hard-to-serve populations along the French Guiana-Suriname border, which showed that the prevalence of these viral infections was somewhat, though not significantly, lower than in urban areas, but still substantial. This study, coupled with the limited knowledge surrounding these viral infections, underscores the need for more accessible and integrated healthcare systems for all populations, including marginalized, undocumented and key populations. Regular screening initiatives, accompanied by awareness campaigns, are crucial for promoting prevention methods such as catch-up HBV vaccination and PrEP. Furthermore, the provision of treatment and care for those already infected remains essential. Lastly, the findings from this study demonstrated that certain cultural practices – such as *bugrus*, vaginal steam baths and *kotis*, – although prevalent, were not associated with HBV, HCV, or HIV infections.

## Supporting information

**S1 File. Supplementary tables.** This file contains the following tables: **S1 Table.** Number of participants and majority ethnicity per site and per country in the MaHeVi study. **S2 Table.** Serological and molecular markers of HBV, HCV, HDV, and HIV infection on dried blood spots in the MaHeVi study. Abbreviations: Ab: antibody, Ag: antigen, HBsAg: hepatitis B surface antigen, HCV: hepatitis C virus, HIV: human immunodeficiency virus, tot Ig Delta: HDV antibodies; ‡ 5/52 positive were later tested on whole blood samples and confirmed negative; † 3/17 positive were later tested on whole blood samples and confirmed negative; * 3/9 equivocal anti-HCV Ab were later tested on whole blood samples and confirmed negative. **S3 Table.** Hepatitis B infection profile according to serological markers in the MaHeVi study. Abbreviations: HBc-Ab: hepatitis B core antibodies, HBs Ab: hepatitis surface antibodies HBsAg: hepatitis B surface antigen, HBV: hepatitis B virus. **S4 Table.** Sub-group analyses for HBV-exposure determinants for men and women separately. Abbreviations: CI: confidence interval, HBV: hepatitis B virus, IQR: inter quartile range, OR: odds ratio, LI: last intercourse, RG: reference group; *Ndjukatongo, Aluku, Pamaka, Saramaka; †Wayana, Teko, Trio, Apalai; All selected known or potential risk factors with a p-value<0.15 are presented in this table; Variables with more than 20% missing data were excluded for the multivariable models (polygamous relationship); Variables with counterintuitive OR (protective effect of piercings and toothbrush sharing) were not included in the multivariate models; ‡ Mother tongue was forced into the multivariable model at first but didn't change the age OR values; Observations with equivocal DBS results were not included in the analyses; P-values in bold are statistically significant. **S5 Table.** Sub-group analyses for HBV-exposure determinants for language communities separately (Amerindian, Maroon, Others). Abbreviations: CI: confidence interval, HBV: hepatitis B virus, IQR: inter quartile range, OR: odds ratio, LI: last intercourse, RG: reference group; *Participants declaring their mother tongue as either Ndjukatongo, Aluku, Pamaka or Saramaka; †Participants declaring their mother tongue as either Wayana, Teko, Trio, Apalai; All selected known or potential risk factors with a p-value<0.15 are presented in this table; Variables with more than 20% missing data were excluded for the multivariable models (polygamous relationship, *bugrus* and transactional sex); Variables with counterintuitive OR (protective effect of piercings and toothbrush sharing) were not included in the multivariate models; ‡ Sex was forced into the multivariable model at first but didn't change the Age OR values; Observations with equivocal DBS results were not included in the analyses; P-values in bold are statistically significant. **S6 Table.** Knowledge, attitudes, practices, and beliefs study results on HBV and HCV in the MaHeVi study. Abbreviations: HBV: hepatitis B virus; HCV: hepatitis C virus, HIV: human immunodeficiency virus; AIDS: acquired immunodeficiency syndrome; Answers on the modes of transmission of viral hepatitis were only considered if the participant declared having already heard of this viral hepatitis.
(XLSX)

**S2 File. Inclusivity in global research questionnaire**
(DOCX)

## Acknowledgments

The authors would like to thank the participants for participating in this study, the community leaders, key informants and members of the focus groups, and the local health care workers on both sides of the river for their valuable insights. The authors would like to thank the MaHevi project team in FG: Valentin Dufit, Dévi Rochemont, Aniza Fahrasmane, William Faurous, Khali Sow, Yann Lambert, Chandrawattie Thibus, Gordon Djani, August Pédié, Chevanna Atooman, Chériline Dinguiou, Franciana Dolianki, Joseph Ouempi, Lita Napuchi, Mélanie Redimoesoe, Meriam Alifons, Itarno Amete, and Louise Hureau, as well as Paul Brousse, Béatrice Pesna, Edouard Legoff, Samuel Gavohedo, Perine Brebion, and all teams from the CDPS, and Amandine Duclau, Emilie Mosnier, and Leila Adriouch; in Suriname: Mariette Jordaan-Kruisland, Elvira Karijoredjo, Deborah Hordijk, Jupta Itoewaki, Mapale (Gio) Itoewaki, Rinia Vyent-Castillon, M. Vyent, L. Truideman, A. Ramcharan, H. Pereira, T. Danso, and as always Edinio Flink.

## Author contributions

**Conceptualization:** Roxane Schaub, M. Sigrid MacDonald–Ottevanger, Edouard Tuaillon, Mathieu Nacher, Stephen Vreden.

**Formal analysis:** Roxane Schaub, Julie Blanc.

**Funding acquisition:** Roxane Schaub, M. Sigrid MacDonald–Ottevanger.

**Investigation:** Roxane Schaub, M. Sigrid MacDonald–Ottevanger, Stella Hoang, Amandine Pisoni, Karine Bolloré, Barbara Biche, Anfernee Neus, Rikesh Bisnajak, Janke Schinkel, Antoon Grunberg, Soeradj Harkisoen, Edouard Tuaillon, Stephen Vreden.

**Methodology:** Roxane Schaub, M. Sigrid MacDonald–Ottevanger, Edouard Tuaillon, Mathieu Nacher, Stephen Vreden.

**Project administration:** Roxane Schaub, M. Sigrid MacDonald–Ottevanger, Stella Hoang.

**Supervision:** Cyril Rousseau, Céline Michaud, Mélanie Gaillet, Emmanuel Gordien, Maria Prins, Edouard Tuaillon, Mathieu Nacher, Stephen Vreden.

**Writing – original draft:** Roxane Schaub, M. Sigrid MacDonald–Ottevanger.

**Writing – review & editing:** Roxane Schaub, M. Sigrid MacDonald–Ottevanger, Stella Hoang, Amandine Pisoni, Karine Bolloré, Barbara Biche, Anfernee Neus, Rikesh Bisnajak, Janke Schinkel, Julie Blanc, Antoon Grunberg, Richard Naldjinan, Aude Lucarelli, Cyril Rousseau, Céline Michaud, Mélanie Gaillet, Soeradj Harkisoen, Emmanuel Gordien, Maria Prins, Edouard Tuaillon, Mathieu Nacher, Stephen Vreden.

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
