## [Decision Letter · Decision Letter 0]

13 Aug 2025

PONE-D-25-23462A cross-border seroprevalence study on HBV, HCV, HDV and HIV in remote Amazonian communities on the border between French Guiana and SurinamePLOS ONE

Dear Dr. Schaub,

Thank you for submitting your manuscript to PLOS ONE. After careful consideration, we feel that it has merit but does not fully meet PLOS ONE’s publication criteria as it currently stands. Therefore, we invite you to submit a revised version of the manuscript that addresses the points raised during the review process.

We look forward to receiving your revised manuscript.

Kind regards,

Getu Girmay

Academic Editor

PLOS ONE

Journal Requirements:

3. We note that Figure 1 in your submission contain map images which may be copyrighted. All PLOS content is published under the Creative Commons Attribution License (CC BY 4.0), which means that the manuscript, images, and Supporting Information files will be freely available online, and any third party is permitted to access, download, copy, distribute, and use these materials in any way, even commercially, with proper attribution. For these reasons, we cannot publish previously copyrighted maps or satellite images created using proprietary data, such as Google software (Google Maps, Street View, and Earth). For more information, see our copyright guidelines: http://journals.plos.org/plosone/s/licenses-and-copyright.

Reviewers' comments:

Reviewer's Responses to Questions

**Comments to the Author**

1. Is the manuscript technically sound, and do the data support the conclusions?

Reviewer #1: Partly

Reviewer #2: Yes

Reviewer #3: Partly

2. Has the statistical analysis been performed appropriately and rigorously? 

Reviewer #1: Yes

Reviewer #2: Yes

Reviewer #3: Yes

3. Have the authors made all data underlying the findings in their manuscript fully available?

Reviewer #1: Yes

Reviewer #2: Yes

Reviewer #3: Yes

4. Is the manuscript presented in an intelligible fashion and written in standard English?

Reviewer #1: Yes

Reviewer #2: Yes

Reviewer #3: Yes

5. Review Comments to the Author

Reviewer #1: The authors performed a seroprevelance study of hepatitis B, C and D and HIV, in the border between French Guiana and Suriname. Some concerns need to be addressed before acceptance of this manuscript

1. The results show an apparent low prevalence of HBsAg in the Amerindian ethnic group. This may be due to the low number of individuals from this ethnic group, in addition to the age of the participants: higher HBsAg prevalence may have been found at lower ages. It would be interesting to include in Table 1 anti-HBc prevalence and perform the analysis of risk factors for HBV according to this marker and not (only) to HBsAg. The presence of anti-HBc is only presented in a supplemental table.

2. Additional Table 2. Footnote is incomplete: some superscript symbols (*, for example), are not explained in the footnote.

3. According to Additional Table 2, the prevalence of anti-HBc is indeed quite high, not reflected in the prevalence of HBsAg. It is important to analyze this (high) prevalence of exposure in each population group and to correlate with the risk factors. This is even more important than the correlation with HBsAg.

4. Did the authors consider to analyze occult HBV infection in at least part of the anti-HBc positive samples?

5. Another concern is the presence of false positive results in these population groups. In rural populations, the presence of polyclonal activation might be frequent, due for example to parasitic infections. This may lead to false positive antibody results, less expected for anti-HBc, which is an inhibition assay (then more specific) but relatively frequent with anti-HCV in some population groups (see for example Rogers KJ, et al., J Clin Lab Anal. 2023 Apr;37(7):e24887. doi: 10.1002/jcla.24887. Epub 2023 Apr 27. The presence of only 20% of viremic samples among the anti-HCV positive ones suggest that some false positive results may have occurred. The authors may consider a second testing with another commercial assay or applying a more stringent criteria of positivity based on signal to cutoff ratio.

6. Discussion. The comparison between anti-HBc prevalence should be included, and limitation on anti-HCV testing.

Reviewer #2: 1. Could you clarify the rationale behind offering an "opt-out" for HIV testing, rather than an "opt-in"?

2. Did the demographic characteristics of the 6% who opted out differ from those who consented, and how might this have affected the final reported HIV prevalence?

3. How was the question about "transactional sex" phrased by the cultural mediators to distinguish it from formally declared "sex work"? Understanding this operational definition is crucial for interpreting its significance as a risk factor.

4. The supporting information notes that some initial positive results from Dried Blood Spots were later found to be negative on whole blood samples. Were the final prevalence figures and risk factor analyses based on the initial screening results or only on the confirmed cases?

5. Could you specify the source and year of the census data used for the post-stratification weighting? Given the high mobility in this border region, how accurate and current is this data considered to be?

6. The literature review is well-focused on the Guiana Shield and South America. However, the discussion could be strengthened by contextualizing the findings with recent literature from other low- and middle-income regions facing similar public health challenges. For example, comparing prevalence rates in key populations (like pregnant women) or vaccination barriers could highlight universal themes. The following recent articles, while geographically distant, explore relevant topics and could be considered for inclusion:

o Seroprevalence of hepatitis B, hepatitis C, and HIV in pregnant women attending a tertiary care hospital in Mogadishu, Somalia, 2017–2021 (DOI: 10.1186/s12879-025-11268-9)

o Prevalence and associated factors for hepatitis B infection among pregnant women attending antenatal clinic at SOS Hospital in Mogadishu, Somalia (DOI: 10.3389/fgwh.2024.1279088)

o Uptake of hepatitis B vaccination and associated factors among health sciences students, Mogadishu, Somalia (DOI: 10.3389/fpubh.2023.1203519)

Reviewer #3: My comments have been attached as a Word document. While the paper presents data with great global health importance, it needs major revision. Specific comments for each section of the paper have been attached. Particularly, the abstract, methods, results, and conclusions sections need more improvement.

6. PLOS authors have the option to publish the peer review history of their article (what does this mean?). If published, this will include your full peer review and any attached files.

Reviewer #1: No

Reviewer #2: No

Reviewer #3: No

---

## [Author Response · Author response to Decision Letter 1]

27 Nov 2025

Authors’ Revisions and Responses to Reviewers’ Comments

We thank the editor and reviewers for their thoughtful and constructive comments. We have carefully revised the manuscript accordingly. Below, we provide a point-by-point response to each comment. Before addressing these comments, we would first like to highlight several changes made to the manuscript that are indirectly related to the reviewer’s comments and important to make following a careful review of the biological results and, in particular, the thresholds used to interpret these results.

We had initially set lower thresholds for the detection of HBsAg and anti-HCV antibodies in order to account for the potential lack of sensitivity of DBS compared to serum from venous blood and to minimize the risk of false negatives, which would have been detrimental to the interests of the participants.

However, for HBsAg, the results that we were subsequently able to verify on venous blood were consistent with the thresholds provided by the manufacturer for the LIAISON® XL Murex HBsAg Quant kit (DiaSorin). We have therefore considered the thresholds for HBsAg as follows, in accordance with the recommendations of the manufacturers of the kits used: reactive (positive) result if index value ≥1.00, non-reactive (negative) result if index value <1.00.

Accordingly, for anti-HCV antibodies, we revised the initially broader cutoffs to bring them closer to the manufacturer’s recommendations for the LIAISON® XL MUREX HCV Ab assay (DiaSorin): non-reactive if S/CO <0.80, equivocal if 0.80–1.00, and reactive if ≥1.00. However, because the assay was performed on DBS eluates, we retained a slightly broader equivocal range, classifying results between >0.20 and <1.00 as equivocal.

This has changed the number of participants in the “HBsAg+” and “HCV+” columns in Tables 1 and 2. We have therefore modified the corresponding numbers in the manuscript text as well as in Additional Tables 2 and 3 (where the modified elements are highlighted in yellow).

Overall, we reviewed all serological cutoff values used in the study and clarified the laboratory procedures in the revised manuscript, specifically in the Materials and Methods section. In addition, detailed information on assay thresholds has been added to Supplementary Table 2.

Reviewer #1:

The authors performed a seroprevalence study of hepatitis B, C and D and HIV, in the border between French Guiana and Suriname. Some concerns need to be addressed before acceptance of this manuscript.

We sincerely thank reviewer #1 for their insightful comments and questions, which we will attempt to address below point by point.

1. The results show an apparent low prevalence of HBsAg in the Amerindian ethnic group. This may be due to the low number of individuals from this ethnic group, in addition to the age of the participants: higher HBsAg prevalence may have been found at lower ages. It would be interesting to include in Table 1 anti-HBc prevalence and perform the analysis of risk factors for HBV according to this marker and not (only) to HBsAg. The presence of anti-HBc is only presented in a supplemental table.

Notably, the vaccination uptake in the Amerindian group was significantly higher than in the non-Amerindian group. Consequently, in the first paragraph of the results, we removed results that were redundant with Table 1 and replaced them with results comparing the sociodemographic characteristics (age, sex, education) of vaccinated and unvaccinated participants.

We agree with the reviewer to include risk factors for HBV exposure based on anti-HBc seropositivity (regardless of HBsAg status) and have added these description to Tables 1 and 2. As the reviewer correctly noted, exploring HBV infection risk factors only in patients with acute or chronic infection (based on HBsAg positivity) was not the most appropriate, as it does not consider infected individuals who have cleared the infection - a significant proportion of those exposed to the virus, particularly adults [Hepatitis B WHO fact sheets]. We therefore revised the HBV risk factor analyses, initially performed only on individuals with positive HBsAg, with an analysis comparing individuals who were HBV exposed to HBV susceptible individuals, which is not only more relevant from a disease history perspective, but also allows us to present much more robust models, due to the larger number of events.

2. Additional Table 2. Footnote is incomplete: some superscript symbols (*, for example), are not explained in the footnote.

We thank the reviewer for their vigilance. We have completed the footnotes for Additional Table 2, corrected “HBe Ab”, which was a typo, to “HBe Ag”, and made changes to the numbers in some cells (highlighted in yellow) following the redefinition of certain thresholds as explained at the beginning of this document:

- we initially indicated “‡ 5/52 positive were later tested on whole blood samples and confirmed negative”, but we removed this sentence because these 5 participants had in fact negative HBs Ag with the new threshold definition we describe at the beginning of the document, and replaced with the following precision “‡ 7/46 positive were later tested on whole blood samples and confirmed positive”

- “† 3/17 [Anti-HCV Ab] positive were later tested on whole blood samples and confirmed negative” was changed to “4 [Anti-HCV Ab] positives were not retested and 1 was already known and treated”

- “* 3/9 equivocal anti-HCV Ab were later tested on whole blood samples and confirmed negative” was changed to “* 4/12 equivocal anti-HCV Ab were later tested on whole blood samples and confirmed negative”.

3. According to Additional Table 2, the prevalence of anti-HBc is indeed quite high, not reflected in the prevalence of HBsAg. It is important to analyze this (high) prevalence of exposure in each population group and to correlate with the risk factors. This is even more important than the correlation with HBsAg.

Please see our response to comment 1. We agree with the reviewer and, in addition to modifying the analysis of HBV infection risk factors to include not only HBsAg-positive participants but also anti-HBc-positive participants, we also performed subgroup analyses by sex and by language community, the detailed results of which have been added in Additional Tables 4 and 5, respectively.

4. Did the authors consider to analyze occult HBV infection in at least part of the anti-HBc positive samples?

This is a very valid point. However, occult hepatitis B infection (OBI) is defined by the persistence of HBV DNA in the absence of detectable HBsAg. By nature, the viral load is extremely low, usually below 200 IU/mL, and often at the level of 10–50 IU/mL, sometimes detectable only intermittently with highly sensitive PCR assays.

Given that the threshold for detection of HBV DNA assays performed on dried blood spots (DBS) is approximately 50-fold higher than that of plasma-based molecular methods, the probability of detecting such low and fluctuating levels of viremia becomes negligible. As a result, DBS testing is not appropriate for the diagnosis or monitoring of OBI, since it would systematically underestimate the true prevalence,

Therefore, HBV DNA testing on DBS cannot be recommended for the evaluation of occult hepatitis B, as it lacks the analytical sensitivity required to capture the very low viral loads that define this condition. Furthermore, we know that anti-HBs is less sensitive on DBS, therefore per WHO recommendations we have included only anti-HB-core positive results as part of resolved HBV infections.

It is not appropriate for diagnosis or monitoring of occult HBV infections (OBI; defined by persistence of HBV DNA in the absence of HBsAg) since HBV loads in case of OBI are generally extremely low, and not detected by molecular DBS testing. Hence, we may have missed OBI.

However, we have added OBI as a limitation to the manuscript: “Fifth, as low level HBV DNA is difficult to detect using DBS, we were unable to collect data on occult HBV infection - defined as presence of HBV DNA in the liver with detectable or undetectable HBV DNA levels in serum, in individuals who are HBsAg seronegative) (EASL 2017 Guidelines). In terms of public health, and prevention of HBV transmission, this aspect warrants further research”.

5. Another concern is the presence of false positive results in these population groups. In rural populations, the presence of polyclonal activation might be frequent, due for example to parasitic infections. This may lead to false positive antibody results, less expected for anti-HBc, which is an inhibition assay (then more specific) but relatively frequent with anti-HCV in some population groups (see for example Rogers KJ, et al., J Clin Lab Anal. 2023 Apr;37(7):e24887. doi: 10.1002/jcla.24887. Epub 2023 Apr 27). The presence of only 20% of viremic samples among the anti-HCV positive ones suggest that some false positive results may have occurred. The authors may consider a second testing with another commercial assay or applying a more stringent criteria of positivity based on signal to cutoff ratio.

We thank reviewer 1 for this comment, which lead us to carefully review our biological results and, in particular, the thresholds used to interpret these results.

As indicated in the introduction to this letter, we had initially set lower thresholds for the detection of anti-HCV antibodies in order to account for the lower sensitivity of DBS compared to serum from venous blood. We were subsequently able to verify on venous blood a few equivocal anti-HCV Ab participant’s result, and they were all confirmed negative. For anti-HCV Ab, we revised the initially broader cutoffs to bring them closer to those indicated by the manufacturer of the LIAISON® XL MUREX HCV Ab kit (DiaSorin): non-reactive (negative) if S/CO value <0.80; equivocal if S/CO value [0.80-1.00], reactive (positive) if S/CO value ≥1.00. After re analyzing the results, we identified five anti HCV positive cases, four of whom also tested positive for HCV RNA which is more consistent with expected HCV serology.

6. Discussion. The comparison between anti-HBc prevalence should be included, and limitation on anti-HCV testing.

We have now added the discussion on anti-HBcore seropositivy. Due to the anti-HCV results being in line with expected HCV serology, we no longer view the anti-HCV testing using DBS as a limitation, and did not add this.

Reviewer #2:

We would like to thank the reviewer for their insightful comments and questions, which we will attempt to address below point by point.

1. Could you clarify the rationale behind offering an "opt-out" for HIV testing, rather than an "opt-in"?

We chose opt-out regimens rather than opt-in for HIV testing, because opt-out can substantially increase HIV testing compared to opt-in schemes (Montoy JCC, Dow WH, Kaplan BC. Patient choice in opt-in, active choice, and opt-out HIV screening: randomized clinical trial. BMJ. 2016 Jan 19;352:h6895; doi: https://doi.org/10.1136/bmj.h6895).

We have added this clarification and the associated reference to the manuscript by replacing the following sentence: “This opt-out was offered because of our perception that HIV testing might be a factor in not participating.” with “We chose the opt-out option for HIV testing based on data indicating this may increase test acceptance rate [ref].”

2. Did the demographic characteristics of the 6% who opted out differ from those who consented, and how might this have affected the final reported HIV prevalence?

We compared the demographic characteristics of the 134 participants who refused the HIV test with the 2152 participants who accepted it. There were no differences in sex, age, professional activity, but there were differences for the following characteristics:

- country of birth: there were less participants who refused HIV testing among those born in Suriname (4.2%) compared to participants born in French Guiana (8.8%) or in another country (5.3%), p=0.002.

- country of residence: there were less participants who refused HIV testing among those living in Suriname (3.1%) compared to participants living in French Guiana (7.5%) or in another country (6.7%), p=0.002.

- school level: HIV testing refusal rate in participants who have been to high school was higher (11.2%) compared to participants who have either never been to school / primary / middle school and higher education, with HIV testing refusal rates of 4.1, 3.8, 6.0 and 6.6%, respectively, p<0.0001.

We have included the following sentence to the manuscript in the results section:

- Six percent (n=134) participants opted out HIV-testing. There were significant differences in country of birth and residence, and educational level between participants who opted out for HIV testing compared to those that did not. There were no differences in sex, age, work, or other determinants.

And in the discussion/limitation section:

- Indeed, 6% opted out for HIV-testing and some of them did so declaring they were HIV-positive. However, the opt-out policy presumably led to a high participation rate; the HBV and HCV prevalence is however, likely more accurate than for HIV. Despite the opt-out, we believe that due to the large sample size, the HIV prevalence estimates are robust.

3. How was the question about "transactional sex" phrased by the cultural mediators to distinguish it from formally declared "sex work"? Understanding this operational definition is crucial for interpreting its significance as a risk factor.

The specific question that the mediators asked was “The last time you had sex, did you pay your partner or were you paid by your partner?”. This precision appears in our manuscript as the following footnote in Table 2: “Transactional sex was defined as having paid for or having been paid in exchange for sex”. This formulation implied that we were asking specifically about money and not gifts, compared to what has been done by other authors, for example Ranganathan et al. in young women in rural South Africa, where they distinguished receiving money and gifts in 2 separate questions.

In our study, because of the design (same questionnaire for both women and men) and setting, we preferred to use a single succinct question, so as not to overload the questionnaire, which could have slowed down inclusions and discouraged some people from participating.

Moreover, this questionnaire was developed based on an extensive preliminary assessment in the communities, including focus group discussions and interviews with key individuals. We believe that the questionnaire was therefore tailored to the population.

However, distinguishing sex work and transactional sex, as well as the nuance between money and gifts is interesting, and it might be worthwhile to explore these issues in a more focused project on the Maroni river, especially among girls and young women.

4. The supporting information notes that some initial positive results from Dried Blood Spots were later found to be negative on whole blood samples. Were the final prevalence figures and risk factor analyses based on the initial screening results or only on the confirmed cases?

As indicated in the introduction to this letter, we had initially set lower thresholds for the detection of HBsAg and anti-HCV antibodies than the thresholds provided by the manufacturer for the LIAISON® XL Murex HBsAg Quant kit (DiaSorin) in order to account for the potential lack of sensitivity of DBS compared to serum from venous blood and to minimize the risk of false negatives, which would have been detrimental to the interests of the participants.

However, for HBsAg, the results that we were subsequently able to verify on venous blood were consistent with the thresholds provided by DiaSorin. We have therefore considered the thresholds for HBsAg as follows, in accordance with the recommendations of the manufacturers of the kits used: reactive (positive) result if index value ≥1.00, non-reactive (negative) result if index value <1.00.

For anti-HCV Ab, the manufacturer of the LIAISON® XL MUREX HCV Ab kit (DiaSorin) indicate the following thresholds: non-reactive (negative) if S/CO value <0.80; equivocal if S/CO value

---

## [Decision Letter · Decision Letter 1]

9 Jan 2026

PONE-D-25-23462R1A cross-border seroprevalence study on HBV, HCV, HDV and HIV in remote Amazonian communities on the border between French Guiana and SurinamePLOS One

Dear Dr. Schaub,

Thank you for submitting your manuscript to PLOS ONE. After careful consideration, we feel that minor revision is mandatory and it does not fully meet PLOS ONE’s publication criteria as it currently stands. Therefore, we invite you to submit a revised version of the manuscript that addresses the points raised during the review process.

We look forward to receiving your revised manuscript.

Kind regards,

Getu Girmay

Academic Editor

PLOS One

Journal Requirements:

Reviewers' comments:

Reviewer's Responses to Questions

**Comments to the Author**

1. If the authors have adequately addressed your comments raised in a previous round of review and you feel that this manuscript is now acceptable for publication, you may indicate that here to bypass the “Comments to the Author” section, enter your conflict of interest statement in the “Confidential to Editor” section, and submit your "Accept" recommendation.

Reviewer #1: All comments have been addressed

Reviewer #3: All comments have been addressed

2. Is the manuscript technically sound, and do the data support the conclusions?

Reviewer #1: Yes

Reviewer #3: Yes

3. Has the statistical analysis been performed appropriately and rigorously? 

Reviewer #1: Yes

Reviewer #3: Yes

4. Have the authors made all data underlying the findings in their manuscript fully available?

Reviewer #1: Yes

Reviewer #3: Yes

5. Is the manuscript presented in an intelligible fashion and written in standard English?

Reviewer #1: Yes

Reviewer #3: Yes

6. Review Comments to the Author

Reviewer #1: The authors addressed satisfactorily the concerns. All the concerns were responded point by point.

Reviewer #3: Dear Authors, thank you for carefully addressing my comments. Most of my comments have been addressed. However, the paper still lacks a separate section from the conclusion, both in the abstract and at the end of the discussion. Conclusion should stand alone as heading in the abstract and body part. If the authors have made subordinate paragraphs of the conclusions section in the discussion, it has to stand independently for readers.

7. PLOS authors have the option to publish the peer review history of their article (what does this mean?). If published, this will include your full peer review and any attached files.

Reviewer #1: No

Reviewer #3: No

---

## [Author Response · Author response to Decision Letter 2]

14 Jan 2026

Authors’ Revisions and Responses to Reviewers’ Comments

First revision:

We thank the editor and reviewers for their thoughtful and constructive comments. We have carefully revised the manuscript accordingly. Below, we provide a point-by-point response to each comment. Before addressing these comments, we would first like to highlight several changes made to the manuscript that are indirectly related to the reviewer’s comments and important to make following a careful review of the biological results and, in particular, the thresholds used to interpret these results.

We had initially set lower thresholds for the detection of HBsAg and anti-HCV antibodies in order to account for the potential lack of sensitivity of DBS compared to serum from venous blood and to minimize the risk of false negatives, which would have been detrimental to the interests of the participants.

However, for HBsAg, the results that we were subsequently able to verify on venous blood were consistent with the thresholds provided by the manufacturer for the LIAISON® XL Murex HBsAg Quant kit (DiaSorin). We have therefore considered the thresholds for HBsAg as follows, in accordance with the recommendations of the manufacturers of the kits used: reactive (positive) result if index value ≥1.00, non-reactive (negative) result if index value <1.00.

Accordingly, for anti-HCV antibodies, we revised the initially broader cutoffs to bring them closer to the manufacturer’s recommendations for the LIAISON® XL MUREX HCV Ab assay (DiaSorin): non-reactive if S/CO <0.80, equivocal if 0.80–1.00, and reactive if ≥1.00. However, because the assay was performed on DBS eluates, we retained a slightly broader equivocal range, classifying results between >0.20 and <1.00 as equivocal.

This has changed the number of participants in the “HBsAg+” and “HCV+” columns in Tables 1 and 2. We have therefore modified the corresponding numbers in the manuscript text as well as in Additional Tables 2 and 3 (where the modified elements are highlighted in yellow).

Overall, we reviewed all serological cutoff values used in the study and clarified the laboratory procedures in the revised manuscript, specifically in the Materials and Methods section. In addition, detailed information on assay thresholds has been added to Supplementary Table 2.

Reviewer #1:

The authors performed a seroprevalence study of hepatitis B, C and D and HIV, in the border between French Guiana and Suriname. Some concerns need to be addressed before acceptance of this manuscript.

We sincerely thank reviewer #1 for their insightful comments and questions, which we will attempt to address below point by point.

1. The results show an apparent low prevalence of HBsAg in the Amerindian ethnic group. This may be due to the low number of individuals from this ethnic group, in addition to the age of the participants: higher HBsAg prevalence may have been found at lower ages. It would be interesting to include in Table 1 anti-HBc prevalence and perform the analysis of risk factors for HBV according to this marker and not (only) to HBsAg. The presence of anti-HBc is only presented in a supplemental table.

Notably, the vaccination uptake in the Amerindian group was significantly higher than in the non-Amerindian group. Consequently, in the first paragraph of the results, we removed results that were redundant with Table 1 and replaced them with results comparing the sociodemographic characteristics (age, sex, education) of vaccinated and unvaccinated participants.

We agree with the reviewer to include risk factors for HBV exposure based on anti-HBc seropositivity (regardless of HBsAg status) and have added these description to Tables 1 and 2. As the reviewer correctly noted, exploring HBV infection risk factors only in patients with acute or chronic infection (based on HBsAg positivity) was not the most appropriate, as it does not consider infected individuals who have cleared the infection - a significant proportion of those exposed to the virus, particularly adults [Hepatitis B WHO fact sheets]. We therefore revised the HBV risk factor analyses, initially performed only on individuals with positive HBsAg, with an analysis comparing individuals who were HBV exposed to HBV susceptible individuals, which is not only more relevant from a disease history perspective, but also allows us to present much more robust models, due to the larger number of events.

2. Additional Table 2. Footnote is incomplete: some superscript symbols (*, for example), are not explained in the footnote.

We thank the reviewer for their vigilance. We have completed the footnotes for Additional Table 2, corrected “HBe Ab”, which was a typo, to “HBe Ag”, and made changes to the numbers in some cells (highlighted in yellow) following the redefinition of certain thresholds as explained at the beginning of this document:

- we initially indicated “‡ 5/52 positive were later tested on whole blood samples and confirmed negative”, but we removed this sentence because these 5 participants had in fact negative HBs Ag with the new threshold definition we describe at the beginning of the document, and replaced with the following precision “‡ 7/46 positive were later tested on whole blood samples and confirmed positive”

- “† 3/17 [Anti-HCV Ab] positive were later tested on whole blood samples and confirmed negative” was changed to “4 [Anti-HCV Ab] positives were not retested and 1 was already known and treated”

- “* 3/9 equivocal anti-HCV Ab were later tested on whole blood samples and confirmed negative” was changed to “* 4/12 equivocal anti-HCV Ab were later tested on whole blood samples and confirmed negative”.

3. According to Additional Table 2, the prevalence of anti-HBc is indeed quite high, not reflected in the prevalence of HBsAg. It is important to analyze this (high) prevalence of exposure in each population group and to correlate with the risk factors. This is even more important than the correlation with HBsAg.

Please see our response to comment 1. We agree with the reviewer and, in addition to modifying the analysis of HBV infection risk factors to include not only HBsAg-positive participants but also anti-HBc-positive participants, we also performed subgroup analyses by sex and by language community, the detailed results of which have been added in Additional Tables 4 and 5, respectively.

4. Did the authors consider to analyze occult HBV infection in at least part of the anti-HBc positive samples?

This is a very valid point. However, occult hepatitis B infection (OBI) is defined by the persistence of HBV DNA in the absence of detectable HBsAg. By nature, the viral load is extremely low, usually below 200 IU/mL, and often at the level of 10–50 IU/mL, sometimes detectable only intermittently with highly sensitive PCR assays.

Given that the threshold for detection of HBV DNA assays performed on dried blood spots (DBS) is approximately 50-fold higher than that of plasma-based molecular methods, the probability of detecting such low and fluctuating levels of viremia becomes negligible. As a result, DBS testing is not appropriate for the diagnosis or monitoring of OBI, since it would systematically underestimate the true prevalence,

Therefore, HBV DNA testing on DBS cannot be recommended for the evaluation of occult hepatitis B, as it lacks the analytical sensitivity required to capture the very low viral loads that define this condition. Furthermore, we know that anti-HBs is less sensitive on DBS, therefore per WHO recommendations we have included only anti-HB-core positive results as part of resolved HBV infections.

It is not appropriate for diagnosis or monitoring of occult HBV infections (OBI; defined by persistence of HBV DNA in the absence of HBsAg) since HBV loads in case of OBI are generally extremely low, and not detected by molecular DBS testing. Hence, we may have missed OBI.

However, we have added OBI as a limitation to the manuscript: “Fifth, as low level HBV DNA is difficult to detect using DBS, we were unable to collect data on occult HBV infection - defined as presence of HBV DNA in the liver with detectable or undetectable HBV DNA levels in serum, in individuals who are HBsAg seronegative) (EASL 2017 Guidelines). In terms of public health, and prevention of HBV transmission, this aspect warrants further research”.

5. Another concern is the presence of false positive results in these population groups. In rural populations, the presence of polyclonal activation might be frequent, due for example to parasitic infections. This may lead to false positive antibody results, less expected for anti-HBc, which is an inhibition assay (then more specific) but relatively frequent with anti-HCV in some population groups (see for example Rogers KJ, et al., J Clin Lab Anal. 2023 Apr;37(7):e24887. doi: 10.1002/jcla.24887. Epub 2023 Apr 27). The presence of only 20% of viremic samples among the anti-HCV positive ones suggest that some false positive results may have occurred. The authors may consider a second testing with another commercial assay or applying a more stringent criteria of positivity based on signal to cutoff ratio.

We thank reviewer 1 for this comment, which lead us to carefully review our biological results and, in particular, the thresholds used to interpret these results.

As indicated in the introduction to this letter, we had initially set lower thresholds for the detection of anti-HCV antibodies in order to account for the lower sensitivity of DBS compared to serum from venous blood. We were subsequently able to verify on venous blood a few equivocal anti-HCV Ab participant’s result, and they were all confirmed negative. For anti-HCV Ab, we revised the initially broader cutoffs to bring them closer to those indicated by the manufacturer of the LIAISON® XL MUREX HCV Ab kit (DiaSorin): non-reactive (negative) if S/CO value <0.80; equivocal if S/CO value [0.80-1.00], reactive (positive) if S/CO value ≥1.00. After re analyzing the results, we identified five anti HCV positive cases, four of whom also tested positive for HCV RNA which is more consistent with expected HCV serology.

6. Discussion. The comparison between anti-HBc prevalence should be included, and limitation on anti-HCV testing.

We have now added the discussion on anti-HBcore seropositivy. Due to the anti-HCV results being in line with expected HCV serology, we no longer view the anti-HCV testing using DBS as a limitation, and did not add this.

Reviewer #2:

We would like to thank the reviewer for their insightful comments and questions, which we will attempt to address below point by point.

1. Could you clarify the rationale behind offering an "opt-out" for HIV testing, rather than an "opt-in"?

We chose opt-out regimens rather than opt-in for HIV testing, because opt-out can substantially increase HIV testing compared to opt-in schemes (Montoy JCC, Dow WH, Kaplan BC. Patient choice in opt-in, active choice, and opt-out HIV screening: randomized clinical trial. BMJ. 2016 Jan 19;352:h6895; doi: https://doi.org/10.1136/bmj.h6895).

We have added this clarification and the associated reference to the manuscript by replacing the following sentence: “This opt-out was offered because of our perception that HIV testing might be a factor in not participating.” with “We chose the opt-out option for HIV testing based on data indicating this may increase test acceptance rate [ref].”

2. Did the demographic characteristics of the 6% who opted out differ from those who consented, and how might this have affected the final reported HIV prevalence?

We compared the demographic characteristics of the 134 participants who refused the HIV test with the 2152 participants who accepted it. There were no differences in sex, age, professional activity, but there were differences for the following characteristics:

- country of birth: there were less participants who refused HIV testing among those born in Suriname (4.2%) compared to participants born in French Guiana (8.8%) or in another country (5.3%), p=0.002.

- country of residence: there were less participants who refused HIV testing among those living in Suriname (3.1%) compared to participants living in French Guiana (7.5%) or in another country (6.7%), p=0.002.

- school level: HIV testing refusal rate in participants who have been to high school was higher (11.2%) compared to participants who have either never been to school / primary / middle school and higher education, with HIV testing refusal rates of 4.1, 3.8, 6.0 and 6.6%, respectively, p<0.0001.

We have included the following sentence to the manuscript in the results section:

- Six percent (n=134) participants opted out HIV-testing. There were significant differences in country of birth and residence, and educational level between participants who opted out for HIV testing compared to those that did not. There were no differences in sex, age, work, or other determinants.

And in the discussion/limitation section:

- Indeed, 6% opted out for HIV-testing and some of them did so declaring they were HIV-positive. However, the opt-out policy presumably led to a high participation rate; the HBV and HCV prevalence is however, likely more accurate than for HIV. Despite the opt-out, we believe that due to the large sample size, the HIV prevalence estimates are robust.

3. How was the question about "transactional sex" phrased by the cultural mediators to distinguish it from formally declared "sex work"? Understanding this operational definition is crucial for interpreting its significance as a risk factor.

The specific question that the mediators asked was “The last time you had sex, did you pay your partner or were you paid by your partner?”. This precision appears in our manuscript as the following footnote in Table 2: “Transactional sex was defined as having paid for or having been paid in exchange for sex”. This formulation implied that we were asking specifically about money and not gifts, compared to what has been done by other authors, for example Ranganathan et al. in young women in rural South Africa, where they distinguished receiving money and gifts in 2 separate questions.

In our study, because of the design (same questionnaire for both women and men) and setting, we preferred to use a single succinct question, so as not to overload the questionnaire, which could have slowed down inclusions and discouraged some people from participating.

Moreover, this questionnaire was developed based on an extensive preliminary assessment in the communities, including focus group discussions and interviews with key individuals. We believe that the questionnaire was therefore tailored to the population.

However, distinguishing sex work and transactional sex, as well as the nuance between money and gifts is interesting, and it might be worthwhile to explore these issues in a more focused project on the Maroni river, especially among girls and young women.

4. The supporting information notes that some initial positive results from Dried Blood Spots were later found to be negative on whole blood samples. Were the final prevalence figures and risk factor analyses based on the initial screening results or only on the confirmed cases?

As indicated in the introduction to this letter, we had initially set lower thresholds for the detection of HBsAg and anti-HCV antibodies than the thresholds provided by the manufacturer for the LIAISON® XL Murex HBsAg Quant kit (DiaSorin) in order to account for the potential lack of sensitivity of DBS compared to serum from venous blood and to minimize the risk of false negatives, which would have been detrimental to the interests of the participants.

However, for HBsAg, the results that we were subsequently able to verify on venous blood were consistent with the thresholds provided by DiaSorin. We have therefore considered the thresholds for HBsAg as follows, in accordance with the recommendations of the manufacturers of the kits used: reactive (positive) result if index value ≥1.00, non-reactive (negative) result if index value <1.00.

For anti-HCV Ab, the manufacturer of the LIAISON® XL MUREX HCV Ab kit (DiaSorin) indicate the following thresholds: non-reactive (negative) if S/CO value <0.80; equivo

---

## [Decision Letter · Decision Letter 2]

18 Feb 2026

A cross-border seroprevalence study on HBV, HCV, HDV and HIV in remote Amazonian communities on the border between French Guiana and Suriname

PONE-D-25-23462R2

Dear Dr. Schaub,

We’re pleased to inform you that your manuscript has been judged scientifically suitable for publication and will be formally accepted for publication once it meets all outstanding technical requirements.

Kind regards,

Getu Girmay

Academic Editor

PLOS One

Reviewers' comments:

Reviewer's Responses to Questions

**Comments to the Author**

1. If the authors have adequately addressed your comments raised in a previous round of review and you feel that this manuscript is now acceptable for publication, you may indicate that here to bypass the “Comments to the Author” section, enter your conflict of interest statement in the “Confidential to Editor” section, and submit your "Accept" recommendation.

Reviewer #1: All comments have been addressed

Reviewer #3: (No Response)

2. Is the manuscript technically sound, and do the data support the conclusions?

Reviewer #1: Yes

Reviewer #3: Yes

3. Has the statistical analysis been performed appropriately and rigorously? 

Reviewer #1: Yes

Reviewer #3: Yes

4. Have the authors made all data underlying the findings in their manuscript fully available?

Reviewer #1: Yes

Reviewer #3: Yes

5. Is the manuscript presented in an intelligible fashion and written in standard English?

Reviewer #1: Yes

Reviewer #3: Yes

6. Review Comments to the Author

Reviewer #1: After revising the response to the concerns and the revised version, I am pleased to inform that all the concerns have been addressed satisfactorely.

Reviewer #3: (No Response)

7. PLOS authors have the option to publish the peer review history of their article (what does this mean?). If published, this will include your full peer review and any attached files.

Reviewer #1: No

Reviewer #3: No

---

## [Editor Report · Acceptance letter]

PONE-D-25-23462R2

PLOS One

Dear Dr. Schaub,

I'm pleased to inform you that your manuscript has been deemed suitable for publication in PLOS One. Congratulations! Your manuscript is now being handed over to our production team.

Kind regards,

on behalf of

Mr. Getu Girmay

Academic Editor

PLOS One